# Episodic Memory-Guided Controllable Experience Synthesis for Reinforcement Learning

Xiao Ma [1 2]   Tian Li [2 3]   Wu-Jun Li [2 3]

## Abstract

In real-world scenarios, data collection for reinforcement learning (RL) is often constrained by safety concerns and high costs, resulting in limited data availability. Diffusion models (DMs) have recently demonstrated remarkable capabilities in capturing complex distributions, making data augmentation a promising approach. However, existing DM-based data augmentation methods still suffer from the limited quality of synthesized data for downstream RL tasks. To overcome this limitation, we propose a novel method called episodic memory-guided controllable experience synthesizer (EMCES). EMCES incorporates an episodic memory-based controllable DM with informative yet concise conditions constructed by episodic memory (EM). To guide the synthesis toward high-quality data, we propose an EM-prioritized condition sampling strategy that leverages EM-based temporal-difference errors to focus generation on data most helpful for RL. Furthermore, we introduce a hashing-based state representation for EM to improve its efficiency and further boost the quality of synthetic data. To the best of our knowledge, EMCES is the first work to incorporate EM into controllable DMs and to leverage EM for guiding data synthesis in RL. Experimental results across multiple environments demonstrate that EMCES significantly improves the quality of the synthetic data, thereby improving the performance of several state-of-the-art RL algorithms. In particular, the hashing-based state representation can reduce storage cost by about 8000-fold and reduce time cost by 25.5-fold, without degrading the normalized score.

[1]School of Computer Science and Technology (School of Artificial Intelligence), Zhejiang Sci-Tech University, P.R.China [2]State Key Lab. for Novel Software Technology, Nanjing University, P.R.China [3]School of Computer Science, Nanjing University, China. Correspondence to: Wu-Jun Li <liwujun@nju.edu.cn>.

*Proceedings of the 43rd International Conference on Machine Learning*, Seoul, South Korea. PMLR 306, 2026. Copyright 2026 by the author(s).

## 1. Introduction

Reinforcement learning (RL) (Sutton & Barto, 2018) has achieved significant progress across a variety of domains, including game playing (Silver et al., 2016), robotics (Miki et al., 2022), and large language models (Ouyang et al., 2022). Due to its trial-and-error learning paradigm, RL typically requires large amounts of data obtained through extensive interactions with the environment to learn effective policies. However, collecting sufficient data in real-world scenarios is frequently constrained by safety concerns and high costs, resulting in a persistent data scarcity problem in RL (Levine et al., 2020). To alleviate this problem, researchers have focused on improving the sample efficiency of RL algorithms (Lin et al., 2018; Mnih et al., 2015; Schaul et al., 2016; Horgan et al., 2018). This problem has also driven the development of offline RL (Levine et al., 2020), where an agent learns a policy from a fixed dataset without direct interactions with the environment. Despite these advancements, data scarcity remains a fundamental challenge in RL. Inspired by the success of data augmentation in computer vision (He et al., 2023b; Trabucco et al., 2024), various data augmentation methods have been introduced into RL, including random augmentation (Laskin et al., 2020; Sinha et al., 2021), generative model-based augmentation methods (Hu et al., 2024; Lu et al., 2023b; Yang & Wang, 2025) and so on. Compared to random augmentation, data synthesized by generative models, particularly diffusion models (DMs), have exhibited higher fidelity (Lu et al., 2023b; Yang & Wang, 2025), meaning they more closely match real-world scenarios. As a result, they have greater potential for improving the performance of RL algorithms.

DM-based augmentation methods can be categorized into two types: trajectory-level and transition-level. Compared with trajectory-level methods (Yang & Wang, 2025; Li et al., 2024; Yang & Wang, 2024; Lee et al., 2024), transition-level methods (Lu et al., 2023b; Wang et al., 2025) are easier to generate high-fidelity data for RL. High-fidelity data implies that the generated transitions faithfully capture the environment dynamics, which is the foundation for effective policy learning. Hence, we focus on the transition-level DM-based augmentation methods. However, the quality of the generated data for policy learning still remains limited.

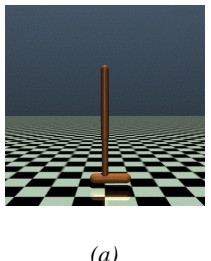

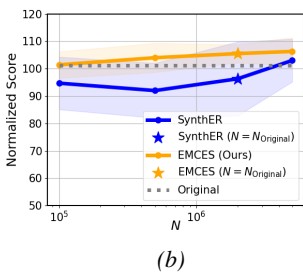

*(a)*          *(b)*

*Figure 1.* (a) A snapshot of Hopper. (b) Results of TD3+BC using the synthetic dataset of varying sizes on Hopper with the 'medium-expert' dataset. Please note that the blue and orange stars indicate the normalized scores based on synthetic data generated by SynthER and EMCES, respectively, with the dataset size aligned with that of the original dataset. The grey dotted line indicates the normalized score using the original dataset.

To demonstrate this limitation more clearly, we evaluate the performance of an agent trained on synthetic datasets using TD3+BC (Fujimoto & Gu, 2021), a classic offline RL algorithm. We conduct experiments in the Hopper environment with the 'medium-expert' dataset, as shown in Fig. 1a. The original 'medium-expert' dataset from the D4RL benchmark (Fu et al., 2020) contains approximately 2M transitions pre-collected directly from the environment. The synthetic datasets are generated by SynthER (Lu et al., 2023b), a classical transition-level DM-based augmentation method for offline and online RL, and range in size from 100K to 5M transitions. Figure 1b presents the normalized scores of policies trained only on synthetic datasets or the original dataset. All results are averaged over five random seeds. We observe that although the synthetic dataset matches the original dataset in size, the performance of the learned policy remains inferior. This indicates that the synthetic dataset lacks sufficient informative transitions to match the quality of the original dataset. Since SynthER (Lu et al., 2023b) does not offer mechanisms to control the quality of the synthetic data, the synthetic dataset must be substantially larger than the original dataset to ensure adequate coverage of high-quality transitions. These observations highlight a fundamental limitation of current transition-level DM-based augmentation methods: the lack of controllability in the synthesis process, which hinders prioritizing high-quality transitions. They also illustrate the necessity of introducing controllable diffusion models (CDMs) (Ho & Salimans, 2021) for transition-level DM-based augmentation methods.

Episodic memory (EM) plays a crucial role in storing and integrating valuable past experiences (Andersen, 2007; Blundell et al., 2016; Lin et al., 2018; Pritzel et al., 2017), providing direct access to high-quality past experiences and thereby improving the sample efficiency of RL algorithms. These valuable experiences stored in EM would provide a possibility for a controllable synthesis process and guide the generation of high-quality data. In this paper, we propose a

novel method called episodic memory-guided controllable experience synthesizer (EMCES) to improve the quality of synthetic data. The main contributions of this work are outlined as follows:

- In EMCES, we propose an EM-based CDM, which leverages EM to construct informative but concise conditions.

- During the data synthesis process, we propose an EM-prioritized condition sampling strategy, which utilizes our proposed EM-based temporal-difference errors to prioritize generating high-quality data by guiding the EM-based CDM.

- We propose a hashing-based state representation for EM, which can improve the efficiency of EM and further boost the quality of synthetic data.

- To the best of our knowledge, EMCES is the first work to embed EM into CDMs and to leverage EM for guiding data synthesis in RL.

- Experimental results across multiple environments demonstrate that EMCES can effectively improve the quality of the synthetic data. In particular, the hashing-based state representation can reduce storage cost by about 8000-fold and reduce time cost by 25.5-fold, without compromising the normalized score.

## 2. Related Work

### 2.1. Diffusion Models in RL

DMs have advanced in RL due to their powerful and flexible distributional modeling ability (Zhu et al., 2023). DMs can be applied across various roles, including policies, planners, and data synthesizers. When used as policies, DMs generate actions conditioned on states (Wang et al., 2023; Chen et al., 2023; Lu et al., 2023c), helping to alleviate issues such as over-conservatism and limited expressiveness in offline RL. When used as planners, DMs combine policy and dynamics modeling to produce multi-step plans or full trajectories (Janner et al., 2022; Zhu et al., 2024; He et al., 2023a; Dong et al., 2024). When used as data synthesizers, DMs can be employed as data augmentation to alleviate data scarcity, as discussed in the following section.

### 2.2. Data Augmentation in RL

Data augmentation in RL can be viewed as a way to alleviate data scarcity. Traditional data augmentation methods expose the agent to multiple views of the same observation to improve the robustness of RL algorithms by adding noise injection or random transformations to raw observations (Laskin et al., 2020; Sinha et al., 2021). In contrast, researchers try

to utilize model-based imaginary samples to augment the offline dataset (Wang et al., 2021; Zhang et al., 2024). Another line of research focuses on synthesizing data using generative models (Lian et al., 2023; Huang et al., 2017), such as variational autoencoders (VAEs) (Kingma & Welling, 2014), generative adversarial networks (GANs) (Goodfellow et al., 2014) and DMs (Ho et al., 2020). In particular, due to the powerful distributional modeling capabilities of DMs (Ho et al., 2020; Karras et al., 2022; Ho & Salimans, 2021), DM-based augmentation methods have recently succeeded in synthesizing high-fidelity data for RL. DM-based augmentation methods can be categorized into two types: trajectory-level and transition-level. Compared with trajectory-level methods (Yang & Wang, 2025; Li et al., 2024; Yang & Wang, 2024; Lee et al., 2024), transition-level methods (Lu et al., 2023b; Huang et al., 2025; Wang et al., 2025) are easier and more flexible to generate high-fidelity data for RL. SynthER (Lu et al., 2023b) is a classical transition-level DM-based augmentation method that can be easily employed in various RL settings, including both offline and online RL. CFDG (Huang et al., 2025) focuses on offline-to-online RL and significantly enhances the generation quality of offline and online data with different distributions. PGR (Wang et al., 2025) only focuses on online RL and uses curiosity or value to guide the data synthesis to improve sample efficiency given an explicit value function. However, existing transition-level data augmentation methods still produce synthesized data of insufficient quality, in the sense that the generated transitions are not sufficiently informative for policy learning, especially in settings without an explicit value function.

## 2.3. Episodic Memory

Humans can rapidly exploit newly discovered high-reward opportunities due to the hippocampus, which stores EM (Squire, 2004; Andersen, 2007). Inspired by this, recent works integrate the EM mechanism into RL to help agents utilize valuable past experiences stored in EM via a non-parametric approach, thereby improving the sample efficiency (Blundell et al., 2016; Pritzel et al., 2017; Lin et al., 2018; Kuznetsov & Filchenkov, 2021; Zheng et al., 2021; Ma & Li, 2023; 2024). Specifically, the EM mechanism maintains a lookup table $\boldsymbol{Q}^{\mathrm{S}}$, where each entry $Q^{\mathrm{S}}(h(s))$ stores a tuple $(h(s), e(h(s)))$, with $h(s)$ serving as the index for lookup. $h(s)$ represents the encoded state of $s$, where $h(\cdot) : \mathcal{S} \to \mathbb{R}^{K}$ is a state representation function, and $K$ denotes the dimension of the encoded state. $e(h(s))$ represents the historical optimal discounted return associated with $h(s)$. Given any trajectory $\tau = \{(s_t, a_t, r_t, s'_t)\}_{t=0}^{T}$, we compute the discounted return $d(s_t) = \sum_{i=t}^{T} \gamma^{i-t} r_i$ at time-step $t \in \{0, \ldots, T\}$, where $s_t$, $a_t$, $r_t$, and $s'_t$ are the corresponding state, action, reward, and next state at time-step $t \in \{0, \ldots, T\}$, respectively. $d(s_t)$ can be abbreviated

as $d_t$. Then, the lookup table $\boldsymbol{Q}^{\mathrm{S}}$ is updated with the set $\{(h(s_t), d_t)\}_{t=0}^{T}$ based on the following rules:

$$Q^{\mathrm{S}}(h(s_t)) \leftarrow \qquad (1)$$
$$\begin{cases} (h(s_t), d_t), & \text{if } Q^{\mathrm{S}}(h(s_t)) \notin \boldsymbol{Q}^{\mathrm{S}}; \\ (h(s_t), e(h(s_t))), & \text{if } Q^{\mathrm{S}}(h(s_t)) \in \boldsymbol{Q}^{\mathrm{S}} \text{ and } e(h(s_t)) \geq d_t; \\ (h(s_t), d_t), & \text{if } Q^{\mathrm{S}}(h(s_t)) \in \boldsymbol{Q}^{\mathrm{S}} \text{ and } e(h(s_t)) < d_t. \end{cases}$$

The specific construction of EM mechanisms follows prior work (Ma & Li, 2023; Kuznetsov & Filchenkov, 2021), and readers can find it in the Appendix. Furthermore, researchers have proposed a class of parametric EM mechanisms (Hu et al., 2021; Ma et al., 2022). However, these parametric EM mechanisms are inherently limited by their reliance on Q-value functions. To the best of our knowledge, SfBC (Chen et al., 2023) is the only work that integrates both the diffusion model and EM mechanism, but it does not utilize the EM mechanism to control CDMs. Hence, our work is the first to introduce the EM mechanism as a control mechanism for CDMs.

### 2.3.1. STATE REPRESENTATION FOR EM

Although state representation has been studied in RL (Schrittwieser et al., 2020; Hafner et al., 2020; Tang et al., 2017), state representation for EM mechanisms is still naive. The default state representation is random projection (RP), a classical dimensionality reduction technique (Kononenko & Kukar, 2007). Each state is encoded as a $K$-dimensional vector of real values. However, real-valued encoded states are unlikely to repeat (Li et al., 2023), leading to inefficient EM updates and high storage, retrieval, and construction costs. Grid-based state representation for EM mechanisms (Li et al., 2023) is proposed to address these issues, where each dimension of states is encoded by discretizing the state space into a grid-like structure. Grid-based state representation groups states that share the same characteristics into a single cluster, enabling more effective EM updates while reducing the complexities of storage, retrieval, and construction. However, the above state representations are both data-independent and ignore the underlying data distribution, leading to inefficient updates in the EM mechanism. These limitations motivate us to design an effective state representation for EM mechanisms.

## 3. Methodology

In this section, we introduce EMCES, a novel method to improve the quality of synthetic data. EMCES consists of three key components: (1) an EM-based CDM constructed by informative yet concise conditions, (2) an EM-prioritized condition sampling strategy to guide the EM-based CDM toward generating high-quality data, and (3) a hashing-based state representation that enhances the efficiency of the EM

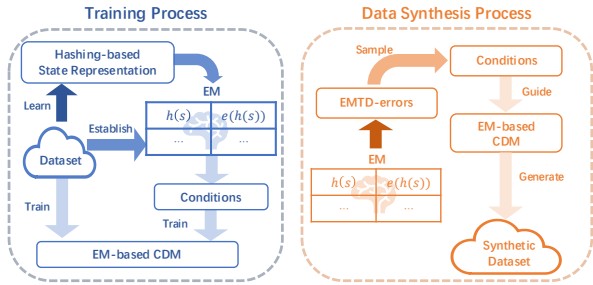

*Figure 2.* Architecture of EMCES during the training and data synthesis processes. Best viewed in color. For each process, darker arrows represent earlier execution steps, while lighter arrows represent later execution steps.

mechanism and further improves synthesis quality. Figure 2 shows the framework of EMCES.

## 3.1. EM-based CDM

A controllable synthesis process is crucial for enhancing the quality of synthetic data, as it enables targeted data generation and improves diversity and relevance for downstream RL tasks. Hence, we introduce CDM (Ho & Salimans, 2021) into our work, and the desired output of CDM is the transition $x(s, a) = (s, a, r, s')$, which represents the fundamental unit of data in RL. To capture the data distribution of the given dataset $D$, we train a CDM to solve the following problem:

$$\min_{\phi} \mathbb{E}_{x \in D, \sigma, \epsilon \sim \mathcal{N}(0, \sigma^2 I)} \|M_{\phi}(x + \epsilon, y(x); \sigma) - x\|_2^2, \quad (2)$$

where $M_{\phi}(x, y; \sigma)$ is a denoiser with parameters $\phi$, $\sigma$ is the noise, and $y(x)$ is the corresponding condition.

The effectiveness of controllable generation heavily relies on the quality of the condition design (Zhang et al., 2023). A well-specified condition $y(x)$ enables the model to produce outputs that align with the intended semantics or dynamics. Hence, the conditions must be carefully designed to encode sufficient information about the target transition, ensuring the generation of desired outputs. To improve training efficiency, the condition should be compact yet informative. Since the state $s$ determines the possible actions, rewards, and subsequent transitions, it must be incorporated into the condition $y(x)$. However, directly employing the raw state representation $s$ might reduce the training efficiency of CDM, as it often contains redundant information, especially in high-dimensional visual states. To address this issue, we adopt a simple yet compact state representation function $h(\cdot)$ to encode the states, and incorporate the encoded representations into the conditioning input. The specific state representation will be described in the following section. While maintaining compactness, the condition should also capture richer contextual information, such as the corresponding action, reward, next state, and potential future

return. To balance informativeness and compactness, we consider using the state-action value function $Q(s, a)$ (or $Q(h(s), a)$) to construct the term $r + \gamma \arg\max_a Q(s', \cdot)$ (or $r + \gamma \arg\max_a Q(h(s'), \cdot)$), which incorporates the corresponding action, reward, next state, and potential future return. Conventional estimation of the state–action value function $Q(s, a)$ relies on neural networks, which demand additional pre-training and often suffer from instability (van Hasselt et al., 2016; Fujimoto et al., 2018; Lin et al., 2018; Ren et al., 2021). Hence, we introduce an EM mechanism to estimate the state–action value function $Q(s, a)$ (or $Q(h(s), a)$), leveraging its non-parametric nature to achieve stable value estimation without the need for extra training (Lin et al., 2018; Zheng et al., 2021; Ma & Li, 2023). To ensure consistency and efficiency, the EM mechanism uses the same state representation function $h(\cdot)$ as that used in the CDM conditions.

In summary, the condition $y(x(s, a), h(\cdot))$ for the transition $x(s, a)$ is defined as follows:

$$y(x(s, a), h(\cdot)) := [h(s), r + \gamma e(h(s'))], \quad (3)$$

where $h(s)$ is the compact state encoding. $r + \gamma e(h(s'))$ denotes the EM-estimated optimal discounted return, which implicitly captures the key elements of a transition and the highest future return potentially by retrieving $e(h(s'))$ from the lookup table $Q^S$ in EM. Hence, this CDM is referred to as the EM-based CDM.

### 3.1.1. TRAINING OF EM-BASED CDM

To obtain the EM-based CDM, we first construct its training dataset. Given a dataset $D$ and a state representation function $h(\cdot)$, we establish an EM mechanism[1]. Based on this, we build a condition set $C = \{y(x(s, a), h(\cdot)) | x(s, a) \in D\}$. The dataset $D$, paired with its corresponding condition set $C$, is then used to optimize the loss function defined in Equation (2). The left part of Figure 2 illustrates the training procedure of the EM-based CDM. After training, the EM-based CDM can serve as an experience synthesizer, generating data conditioned on $y(x(s, a), h(\cdot))$. For sampling, we employ the EDM method (Karras et al., 2022).

## 3.2. EM-prioritized Condition Sampling

Although the EM-based CDM can be used directly as an experience synthesizer, its primary strength lies in generating high-quality data for RL algorithms. Intuitively, the data synthesis process should not only adhere to the underlying data distribution but also prioritize the generation of data that maximally benefits the agent's training. While generating high-return transitions may intuitively boost performance (Liu et al., 2024), this approach often yields brittle

---

[1]Please note that the establishment of the EM mechanism can be found in the Appendix.

policies (Schaul et al., 2016), especially when trained on sub-optimal datasets. Prioritizing informative transitions has been shown to enhance both sample efficiency and stability in off-policy learning (Schaul et al., 2016; Fan et al., 2025; Wang et al., 2017). In particular, transitions with large temporal-difference (TD) errors are recognized as highly valuable, as they highlight regions where the Q-function has space for improvement (Schaul et al., 2016).

Motivated by this insight, we propose an EM-based TD error (EMTD-error) and an EM-prioritized condition sampling strategy. EMTD-error, denoted as $\delta(x(s,a), h(\cdot))$, serves as a principled measure of the importance of the transition $x(s,a)$ for policy improvement and is defined as follows:

$$\delta(x(s,a), h(\cdot)) = r + \gamma e(h(s')) - e(h(s)), \quad (4)$$

where $e(h(s))$ and $e(h(s'))$ are retrieved from $\boldsymbol{Q}^{\mathrm{S}}$ by indexing $h(s)$ and $h(s')$, respectively. The magnitude of the EMTD-error indicates how far the value based on the next state $r + \gamma e(h(s'))$ deviates from the historical optimal discounted return $e(h(s))$ of the current state $s$. Hence, the EMTD-error serves as an indicator of the potential utility of the transition for policy improvement. A higher EMTD-error indicates that the transition could yield greater returns than previously observed ones from the same state representation, making it more informative for policy improvement. To avoid over-sampling a small subset of transitions with high EMTD-errors, we propose an EM-prioritized condition sampling strategy to balance informativeness with diversity. Specifically, we apply a softmax operator over the EMTD-errors $\delta(x(s,a), h(\cdot))$ to compute a sampling probability for each condition $y(x(s,a), h(\cdot))$, shown as follows:

$$p(y(x(s,a), h(\cdot))) = \frac{\exp^{\beta\delta(x(s,a), h(\cdot))}}{\sum_{x(s,a)\in D} \exp^{\beta\delta(x(s,a), h(\cdot))}}, \quad (5)$$

where $\beta \in [0,1]$ determines the level of prioritization. A higher value of $\beta$ increases the emphasis on transitions with larger EMTD-errors, while $\beta = 0$ corresponds to uniform sampling. Hence, based on this EM-prioritized condition sampling strategy, we guide the EM-based CDM to generate high-quality data while preserving diversity, ensuring robust policy learning.

### 3.3. Hashing-based State Representation

A well-designed state representation facilitates the construction of an effective EM mechanism by aggregating states across different trajectories and reducing both storage and time costs. In the context of EM-based CDM, it also facilitates the construction of compact yet informative conditions, which are essential for effective training and guiding the synthesis of high-quality data. However, existing state representation methods for EM are data-independent, which

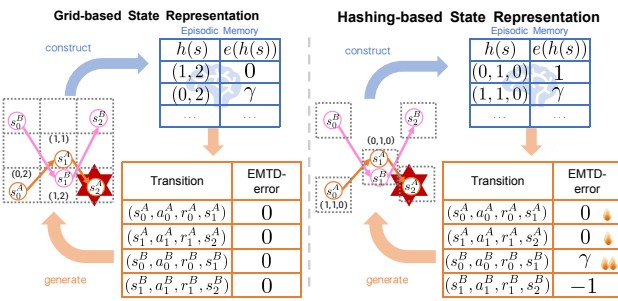

*Figure 3.* Illustrative example of different state representations with two trajectories $\tau^A = (s_0^A, a_0^A, r_0^A, s_1^A, a_1^A, r_1^A, s_2^A)$ and $\tau^B = (s_0^B, a_0^B, r_0^B, s_1^B, a_1^B, r_1^B, s_2^B)$. Each circle and each grey box represents a state and a code, respectively. Agent in the area of the red star receives a reward of 1; otherwise, 0.

limits the efficiency of the EM mechanism. These limitations motivated us to design a novel state representation for the EM mechanism.

To improve the efficiency of the EM mechanism, we introduce a hashing-based state representation that leverages a learning-to-hash approach to encode states with dimension of $F$ into compact and informative binary codes. Specifically, it learns $K \leq F$ real-valued projection functions $f(s) = \{f_k(s)\}_{k=1}^{K}$ given the dataset $D$, where each function outputs a real value and $K$ denotes the length of the hash codes. Each projected dimension is then quantized into one bit using a hash function $h_k(s) = \mathrm{sgn}(f_k(s)), k \in \{1, \ldots, K\}$, where $\mathrm{sgn}(x) = 1$ if $x \geq 0$, and 0 otherwise. We adopt IsoHash (Kong & Li, 2012), a state-of-the-art unsupervised learning-to-hash approach, to learn these projection functions. For details about IsoHash, please refer to the original paper (Kong & Li, 2012). Because hash codes are learned from the data distribution, they can align with the environment's underlying structure, reducing aliasing between unrelated states. Furthermore, the hashing-based state representation attributes similar states with the same hash code, as shown in Figure 3, which is helpful in constructing a more effective EM mechanism by implicitly merging multiple trajectories. This state representation also provides compact and sufficient conditions for an effective EM-based CDM. Please note that the information retrieved from the EM mechanism is not used directly; instead, it is combined with the current transition's state and action (e.g., Equation (3) and Equation (4)). This design ensures that our method remains effective even in state-sensitive tasks.

#### 3.3.1. COMPLEXITY ANALYSIS

For the EM mechanisms, its corresponding storage[2], retrieval time, and construction time complexities are

---

[2]In this paper, we only focus on the storage complexity of encoded states in EM mechanisms.

*Table 1.* Overall results on D4RL environments using selected state-of-the-art offline RL algorithms. Note that for each task, all methods are evaluated on datasets of equal size.

| Environment | Task | TD3+BC | | | IQL | | | EDAC | | |
|---|---|---|---|---|---|---|---|---|---|---|
| | | Original | SynthER | EMCES | Original | SynthER | EMCES | Original | SynthER | EMCES |
| Half Cheetah | random | $11.4_{\pm1.2}$ | $11.5_{\pm0.6}$ | $12.5_{\pm0.4}$ | $9.3_{\pm4.2}$ | $13.8_{\pm1.2}$ | $13.7_{\pm1.8}$ | $17.2_{\pm10.5}$ | $16.5_{\pm5.9}$ | $18.6_{\pm1.2}$ |
| | medrep | $44.8_{\pm0.6}$ | $44.5_{\pm0.7}$ | $45.2_{\pm0.6}$ | $44.5_{\pm0.2}$ | $44.3_{\pm0.3}$ | $44.3_{\pm0.2}$ | $64.6_{\pm2.1}$ | $55.8_{\pm2.4}$ | $56.5_{\pm0.5}$ |
| | medium | $48.1_{\pm0.2}$ | $48.7_{\pm0.3}$ | $48.8_{\pm0.2}$ | $48.3_{\pm0.2}$ | $48.9_{\pm0.3}$ | $49.3_{\pm0.1}$ | $67.5_{\pm1.0}$ | $63.9_{\pm2.5}$ | $68.2_{\pm1.0}$ |
| | medexp | $90.8_{\pm6.0}$ | $76.2_{\pm13.4}$ | $90.7_{\pm3.3}$ | $94.8_{\pm0.5}$ | $92.1_{\pm5.2}$ | $93.8_{\pm1.2}$ | $79.7_{\pm9.5}$ | $63.7_{\pm7.6}$ | $75.8_{\pm10.0}$ |
| Walker2d | random | $0.4_{\pm0.3}$ | $1.8_{\pm2.4}$ | $4.0_{\pm2.5}$ | $3.2_{\pm1.1}$ | $3.4_{\pm1.1}$ | $3.7_{\pm0.3}$ | $7.0_{\pm6.9}$ | $10.8_{\pm8.8}$ | $21.6_{\pm1.1}$ |
| | medrep | $85.6_{\pm4.0}$ | $85.9_{\pm1.8}$ | $85.7_{\pm2.3}$ | $82.2_{\pm3.0}$ | $82.0_{\pm8.8}$ | $82.4_{\pm2.1}$ | $83.1_{\pm1.3}$ | $74.9_{\pm1.4}$ | $72.6_{\pm0.3}$ |
| | medium | $82.7_{\pm4.8}$ | $84.1_{\pm0.3}$ | $84.2_{\pm1.1}$ | $80.9_{\pm3.2}$ | $83.0_{\pm4.0}$ | $86.1_{\pm2.1}$ | $90.4_{\pm1.6}$ | $88.2_{\pm1.0}$ | $89.4_{\pm1.0}$ |
| | medexp | $110.0_{\pm0.4}$ | $110.5_{\pm0.5}$ | $110.7_{\pm0.3}$ | $111.7_{\pm0.9}$ | $111.2_{\pm1.0}$ | $111.7_{\pm0.5}$ | $113.5_{\pm0.5}$ | $111.6_{\pm1.0}$ | $112.9_{\pm0.6}$ |
| Hopper | random | $8.7_{\pm0.5}$ | $7.8_{\pm0.6}$ | $14.1_{\pm8.7}$ | $7.3_{\pm0.3}$ | $7.9_{\pm1.0}$ | $7.4_{\pm0.2}$ | $6.2_{\pm1.6}$ | $5.2_{\pm1.1}$ | $16.4_{\pm12.2}$ |
| | medrep | $64.4_{\pm21.5}$ | $58.9_{\pm14.0}$ | $68.5_{\pm14.1}$ | $97.4_{\pm6.4}$ | $42.3_{\pm37.9}$ | $101.0_{\pm0.9}$ | $100.1_{\pm0.9}$ | $101.0_{\pm0.7}$ | $99.6_{\pm0.8}$ |
| | medium | $60.4_{\pm3.5}$ | $57.3_{\pm2.6}$ | $61.5_{\pm2.8}$ | $67.6_{\pm3.8}$ | $65.9_{\pm2.3}$ | $67.4_{\pm2.7}$ | $101.6_{\pm1.2}$ | $100.0_{\pm4.3}$ | $100.4_{\pm1.4}$ |
| | medexp | $101.2_{\pm9.1}$ | $96.3_{\pm13.5}$ | $105.5_{\pm3.3}$ | $107.4_{\pm7.8}$ | $85.2_{\pm27.0}$ | $102.9_{\pm8.8}$ | $106.7_{\pm4.0}$ | $99.8_{\pm14.8}$ | $105.0_{\pm7.0}$ |
| **sum** | | 708.5 | 683.5 | **731.4** | 754.6 | 680.0 | **763.7** | **837.7** | 791.4 | 837.0 |
| maze2d | umaze | $29.4_{\pm12.3}$ | $37.0_{\pm26.8}$ | $41.9_{\pm3.1}$ | $42.1_{\pm0.6}$ | $40.4_{\pm2.4}$ | $40.4_{\pm1.2}$ | $104.8_{\pm21.9}$ | $92.1_{\pm11.0}$ | $92.2_{\pm3.6}$ |
| | medium | $59.5_{\pm36.3}$ | $50.8_{\pm32.8}$ | $87.7_{\pm47.3}$ | $34.9_{\pm2.7}$ | $35.0_{\pm1.7}$ | $35.1_{\pm0.7}$ | $62.7_{\pm8.5}$ | $61.7_{\pm16.1}$ | $95.6_{\pm27.9}$ |
| | large | $97.1_{\pm25.4}$ | $105.5_{\pm52.3}$ | $102.3_{\pm44.7}$ | $61.7_{\pm3.5}$ | $59.4_{\pm6.3}$ | $66.1_{\pm2.2}$ | $134.2_{\pm32.1}$ | $123.2_{\pm25.2}$ | $144.7_{\pm20.2}$ |
| **sum** | | 186.0 | 193.3 | **231.9** | 138.7 | 134.8 | **141.6** | 301.7 | 277.0 | **332.5** |

$\mathcal{O}(cKn)$, $\mathcal{O}(\log n)$ and $\mathcal{O}(n \log n)$, respectively. Here, $n$ denotes the number of distinguished encoded states given a dataset $D$ of size $N$, and $c > 0$ is the number of bits required per dimension, and $K$ denotes the dimension of encoded states. The subscripts of $n$ and $c$ refer to the specific state representation. Following prior works (Blundell et al., 2016; Lin et al., 2018; Ma & Li, 2023), the lookup tables in EM mechanisms are all implemented using the KD-tree data structure (Bentley, 1975). With a fixed $K$, $n_{\text{RP}}$ is typically on the same order as $N$, while $n_{\text{Grid}} \ll N$ and $n_{\text{Hash}} \ll N$. Hence, the hashing-based state representation can reduce the retrieval time and construction time complexities compared with the RP-based state representation. Moreover, because $c_{\text{Hash}} = 1$ in our hashing-based state representation, whereas $c_{\text{RP}} = 32^3$ and $c_{\text{Grid}} \geq 1$, the hashing-based state representation yields a notable reduction in storage complexity.

### 3.4. Adaptive to Different RL Settings

During the training process, given a dataset $D$, we first learn a hashing-based state representation and then construct the EM mechanism accordingly. This EM mechanism is used to construct conditions and then build an EM-based CDM. During the data synthesis process, we use EM-prioritized condition sampling to guide the EM-based CDM in generating high-quality data. In offline RL, synthetic data can either entirely replace the original dataset or upsample it. In online RL, as the original dataset collected from the envi-

ronment is continuously populated, the hashing-based state representation, EM mechanism, and EM-based CDM need to be periodically updated. The agent then updates its policy using the dataset augmented with the synthetic data. In summary, EMCES is adaptive to both offline and online RL settings.

## 4. Experiments

In this section, we verify the effectiveness of EMCES across different RL settings to demonstrate its ability to synthesize high-quality data. Especially in the offline RL setting, we show that our method can synthesize high-quality data when scaling to vision-based environments. We conduct ablation studies to validate the effectiveness of different components in EMCES. Please note that all results in this paper are summarized over five random seeds.

### 4.1. Main Results

#### 4.1.1. OFFLINE RL

Following prior work (Lu et al., 2023b), we first consider four proprioceptive environments from D4RL (Fu et al., 2020): HalfCheetah, Walker2d, Hopper, and Maze2D. For comparison, we select three classic state-of-the-art offline RL algorithms, including TD3+BC (Fujimoto & Gu, 2021), IQL (Kostrikov et al., 2022) and EDAC (An et al., 2021). We select SynthER as the baseline, and the performance trained by the original datasets from D4RL serves as another baseline. Additionally, we include experimental results of RTDiff (Yang & Wang, 2025), which are provided in the

---

[3] In the RP-based state representation, each dimension is stored as a 32-bit single-precision floating-point value.

*Table 2.* Results of all methods on VD4RL environments using BC and DrQ+BC, where each result denotes the mean episode reward and standard deviation.

| Environment | | BC | | | DrQ+BC | | |
|---|---|---|---|---|---|---|---|
| | | Original | SynthER | EMCES | Original | SynthER | EMCES |
| *walker_walk* | expert | 879.18 ± 36.52 | 901.71 ± 38.64 | **947.43 ± 10.21** | 399.84 ± 108.88 | **717.25 ± 29.49** | 708.67 ± 24.88 |
| | medrep | **121.61 ± 27.01** | 48.79 ± 13.60 | 93.48 ± 14.97 | 169.19 ± 120.74 | 230.35 ± 60.54 | **289.77 ± 32.41** |
| | medium | 365.11 ± 51.93 | 321.81 ± 37.64 | **391.91 ± 9.67** | 457.05 ± 15.59 | 447.62 ± 39.69 | **491.59 ± 10.52** |
| | medexp | 448.13 ± 49.02 | 389.16 ± 28.92 | **458.92 ± 53.80** | 725.38 ± 79.87 | 686.36 ± 81.28 | **790.18 ± 37.96** |
| *cheetah_run* | expert | 621.23 ± 72.80 | 820.34 ± 29.78 | **857.97 ± 7.96** | 253.47 ± 32.62 | 266.53 ± 28.04 | **273.96 ± 23.99** |
| | medrep | **126.44 ± 52.49** | 27.75 ± 19.29 | 93.48 ± 14.97 | 448.69 ± 10.17 | 468.59 ± 9.15 | **475.44 ± 3.29** |
| | medium | 451.63 ± 27.54 | 505.40 ± 11.87 | **522.79 ± 4.00** | 512.12 ± 25.69 | 561.35 ± 6.22 | **573.27 ± 4.29** |
| | medexp | 507.24 ± 19.99 | 519.78 ± 5.73 | **536.60 ± 31.58** | 350.29 ± 73.04 | 401.02 ± 41.28 | **482.72 ± 44.05** |
| **average** | | 440.08 | 441.84 | **487.82** | 414.50 | 472.38 | **518.00** |

Appendix due to space constraints. For a fair evaluation of data quality, the agent is trained on synthetic datasets generated by data augmentation methods, each of which matches the size of the original dataset. The resulting performances are reported as normalized scores across all tasks. Detailed hyperparameter settings and the rationale behind their selection are provided in the Appendix, along with the results of hyperparameter sensitivity analysis for $K$ and $\beta$. As shown in Table 1, our method consistently improves the performance of offline RL algorithms across nearly all tasks and often matches or exceeds training on the original datasets, indicating higher-quality synthetic transitions. Furthermore, to illustrate the high-fidelity of synthesized data by EMCES, we measure the faithfulness of generated transitions to environment dynamics in the Appendix, following prior work (Lu et al., 2023b; Wang et al., 2025).

We further scale our method to vision-based environments from the VD4RL (Lu et al., 2023a) benchmark, where the original image observations are of size $84 \times 84 \times 3$. We focus on the 'cheetah-run' and 'walker-walk' environments. Following prior work (Lu et al., 2023b), image observations are encoded into a latent space using a pretrained CNN. All data augmentation methods upsample 5M transitions in the latent space. We choose BC (Lu et al., 2023a) and DrQ+BC (Lu et al., 2023a) as the RL algorithms. As shown in Table 2, EMCES consistently outperforms both the original datasets and SynthER under BC and DrQ+BC.

### 4.1.2. ONLINE RL

To further demonstrate the robustness and general applicability of EMCES, we also validate its effectiveness in online RL. We select six environments from the DMC suite (Tunyasuvunakool et al., 2020) and the OpenAI Gym Suite (Brockman et al., 2016) to evaluate our method, including quadruped-walk, reacher-hard, cheetah-run, Walker2d, HalfCheetah, and Hopper. We use SAC (Haarnoja et al., 2018) as the RL algorithm to evaluate data augmenta-

tion methods, as it is a classical online RL algorithm. REDQ (Chen et al., 2021), a sample-efficient variant of SAC, serves as an additional baseline. In addition to SynthER (Lu et al., 2023b), we also compare against PGR (Wang et al., 2025), a recent data augmentation method that focuses only on online RL. We denote SAC combined with EMCES, SynthER, and PGR as SAC (EMCES), SAC (SynthER), and SAC (PGR), respectively. All methods are evaluated over 200K online steps for DMC environments and 100K online steps for OpenAI Gym environments. Additional experimental details can be found in the Appendix. The training curves of all methods on six environments are shown in Figure 4. SAC (EMCES) consistently improves sample efficiency and outperforms SAC (SynthER) and SAC (PGR), demonstrating that the synthetic data generated by EMCES is of higher quality.

### 4.2. Ablation Studies

To investigate the contribution of each component in our method EMCES, we perform a series of ablation studies on the Hopper environment with the 'medexp' dataset, employing TD3+BC as the RL algorithm.

**Effect of the conditions** The design of conditions is important for controlling the synthesis process in EMCES, as it dictates how the CDM captures the underlying dynamics of transitions. To show the effect of the conditions, we compare our method with alternative types of conditions, including $[h(s)]$, $[r + \gamma e(h(s'))]$, and $[h(s), d]$. Table 3 shows that using $[h(s)]$ and $[r + \gamma e(h(s'))]$ simultaneously leads to better performance compared to using either one alone. This indicates that combining $[h(s)]$ and $[r + \gamma e(h(s'))]$ as the conditions is necessary. Furthermore, replacing $r + \gamma e(h(s'))$ with the discounted return $d(s)$ results in decreased performance, although it still surpasses SynthER. This illustrates that although the discounted return $d(s)$ also contains the cumulative information, the structure of $r + \gamma e(h(s'))$ aligns more closely with the state-action

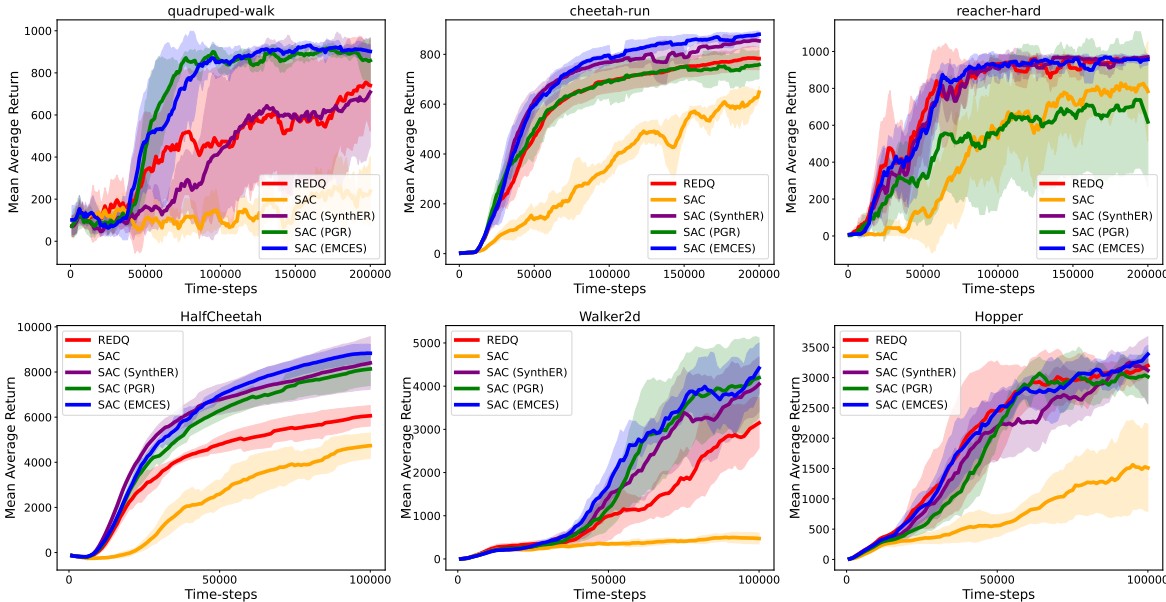

*Figure 4.* Training curves of SAC (EMCES), SAC (SynthER), SAC (PGR), SAC and REDQ. The solid line represents the mean average return, and the shaded area represents one standard deviation.

value function, making it a more sufficient and effective condition for transition modeling. These results all validate that using $[h(s), r + \gamma e(h(s'))]$ as the condition can sufficiently reflect the key characteristics of transitions, thereby facilitating effective policy learning.

**Effect of the condition sampling strategies** As for the EM-prioritized condition sampling strategy, we compare it with historical optimal discounted return-based, uniform-based, return-based, and reward-based condition sampling strategies. Specifically, historical optimal discounted return-based, return-based, reward-based condition sampling strategies use $e(h(s))$, $R$, and $r$ to replace $\delta(x(s, a), h(\cdot))$ in Equation (5), respectively. Uniform condition sampling is equivalent to the EM-prioritized condition sampling strategy when $\beta = 0$. As shown in Table 3, the EM-prioritized condition sampling strategy can improve the performance by guiding the synthesis of data with high EMTD-errors, surpassing other sampling strategies. These results verify the necessity of the EM-prioritized condition sampling strategy.

**Effect of the state representations** The choice of state representation decides the efficiency of EM, which further influences the EM-based CDM and the quality of synthesized data. Hence, we summarize the normalized scores of EMCES under different state representations in Table 4. To ensure a fair comparison with hashing-based state representation, $b_{\text{Grid}}$ is set to 1. As shown in Table 4, EMCES with different state representations all achieve better performance than baselines, especially SynthER. This verifies

*Table 3.* Performance of EMCES with different conditions and condition sampling strategies on Hopper with the 'medexp' dataset. Numbers in parentheses indicate performance improvements relative to SynthER.

| Elements in Condition | | Condition Sampling Strategy | Normalized Score |
|---|---|---|---|
| State | State-Action Value | | |
| EMCES (Our method) | | | |
| $h(s)$ | $r + \gamma e(h(s'))$ | EMTD-error | **105.5 ± 3.3** (+9.2) |
| $h(s)$ | - | EMTD-error | 91.7 ± 11.5 (-4.6) |
| - | $r + \gamma e(h(s'))$ | EMTD-error | 91.0 ± 15.9 (-5.3) |
| $h(s)$ | $d(s)$ | EMTD-error | 100.9 ± 8.8 (+4.6) |
| $h(s)$ | $r + \gamma e(h(s'))$ | $e(h(s))$ | 103.6 ± 6.1 (+7.4) |
| $h(s)$ | $r + \gamma e(h(s'))$ | Uniform | 91.6 ± 7.5 (-4.7) |
| $h(s)$ | $r + \gamma e(h(s'))$ | Return | 95.2 ± 5.2 (-1.1) |
| $h(s)$ | $r + \gamma e(h(s'))$ | Reward | 99.4 ± 9.8 (+3.1) |

the effectiveness of the framework of our method EMCES. The hashing-based state representation performs best among all state representations. Here, we conduct experiments on a workstation equipped with a 36-core, 72-thread Intel(R) Xeon(R) Gold 6240 CPU @ 2.60GHz, 377 GB RAM, and 8 NVIDIA GeForce RTX 2080 Ti GPUs. Table 4 also reports the storage and time costs (including retrieval and construction time) required for establishing the EM mechanism. Hashing-based and grid-based state representations can both significantly reduce the storage and time costs. Compared to the RP-based state representation, the hashing-based state representation reduces storage and time costs by approximately 8000-fold and 25.5-fold, respectively. Compared to the grid-based state representation, the hashing-based state representation can help the EM-based CDM gener-

*Table 4.* Results of EMCES with different state representations (SRs) on Hopper with the 'medexp' dataset. The best results are highlighted in bold, while the second-best results are underlined. The number in brackets denotes the performance increase by the variants of EMCES compared to the result achieved by SynthER.

| SR | Storage Cost (KB) | Time Cost (s) | Normalized Score |
| --- | --- | --- | --- |
| RP | 7806.49 | 7573.88 | $\underline{104.0 \pm 8.1}$ (+7.7) |
| Grid | **0.86** | **293.12** | $98.2 \pm 12.3$ (+1.9) |
| Hashing (Ours) | $\underline{0.96}$ | $\underline{297.12}$ | $\mathbf{105.5 \pm 3.3}$ (+9.2) |

ate higher-quality data to boost the agent's performance. These findings validate that our proposed hashing-based state representation is essential for guiding the synthesis of high-quality data.

## 5. Conclusion

In this paper, we propose a novel and effective method, EMCES, to enhance the quality of synthetic data. In EM-CES, we propose an EM-based CDM, where the EM mechanism is utilized to construct informative and concise conditions. During the data synthesis phase, we propose an EM-prioritized condition sampling strategy to guide the EM-based CDM to synthesize high-quality data. We also propose a hashing-based state representation to improve the efficiency of the EM mechanism and the quality of synthetic data. EMCES is the first work to introduce EM for controlling and further guiding data synthesis. Experimental results show that EMCES can effectively improve the quality of synthetic data.

## Acknowledgements

This work was supported by the NSFC (Grant No. 62506341), the Zhejiang Provincial Natural Science Foundation of China (Grant No. LQN26F020034), the Science Foundation of Zhejiang Sci-Tech University (Grant No. 25232128-Y), the Fundamental Research Funds for the Central Universities and Nanjing University International Collaboration Initiative (Grant No. 020214380129), Fundamental and Interdisciplinary Disciplines Breakthrough Plan of the Ministry of Education of China (Grant No. JYB2025XDXM118), and "111 Center" (Grant No. B26023).

## Impact Statement

Real-world RL often suffers from limited data due to high interaction costs and safety constraints, which hinder the deployment of RL agents in practical applications. EMCES addresses this challenge by leveraging episodic memory to guide controllable diffusion models for high-quality data synthesis, significantly improving policy learning efficiency. The broader impact of this research includes reducing the need for expensive online interactions and making RL more feasible in interaction-sensitive domains such as robotics, autonomous vehicles, and industrial control. From the ethical perspective, our method aims to improve the quality of data synthesis for policy learning and does not inherently introduce bias. Nevertheless, in high-risk scenarios, caution is required to avoid excessive dependence on synthetic data without adequate validation. Overall, this research has the potential to accelerate the adoption of RL in safety-critical and cost-sensitive applications while contributing to responsible and data-efficient machine learning.

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

# A. Background

## A.1. Episodic Memory

Humans can rapidly exploit newly discovered high-reward opportunities due to the hippocampus, which stores episodic memory (EM) (Squire, 2004; Andersen, 2007). Inspired by this, recent works integrate the EM mechanism into RL to help agents utilize valuable past experiences stored in EM through a non-parametric approach, thereby improving the sample efficiency (Blundell et al., 2016; Pritzel et al., 2017; Lin et al., 2018; Kuznetsov & Filchenkov, 2021; Zheng et al., 2021; Ma & Li, 2023). EM mechanisms can be categorized into state-action-based (Blundell et al., 2016; Pritzel et al., 2017; Lin et al., 2018) and state-based EM mechanisms (Ma & Li, 2023; Yang et al., 2023). In particular, the state-based EM mechanism has not only proven its effectiveness but also demonstrated its high computational efficiency. Hence, we adopt the state-based EM mechanism in this paper. Specifically, the EM mechanism maintains a lookup table $\boldsymbol{Q}^{\mathrm{S}}$, where each entry $Q^{\mathrm{S}}(h(s))$ stores a tuple $(h(s), e(h(s)))$, with $h(s)$ serving as the index for lookup. Please note that $h(s)$ represents the encoded state of $s$, where $h(\cdot) : \mathcal{S} \to \mathbb{R}^K$ is a state representation function, and $K$ denotes the dimension of the encoded state. $e(h(s))$ stores the historical optimal discounted return associated with $h(s)$. The EM mechanism has two operators: lookup and update. The lookup operator retrieves $e(h(s))$, the historical optimal discounted return for a given state $s$, by using $h(s)$ as an index in the lookup table $\boldsymbol{Q}^{\mathrm{S}}$. Given any trajectory $\tau = \{(s_t, a_t, r_t, s_t')\}_{t=0}^T$, we compute the discounted return $d(s_t) = \sum_{i=t}^T \gamma^{i-t} r_i$ at time-step $t \in \{0, \ldots, T\}$, where $s_t$, $a_t$, $r_t$, and $s_t'$ are the corresponding state, action, reward, and next state at time-step $t \in \{0, \ldots, T\}$, respectively. $d(s_t)$ can be abbreviated as $d_t$. And then the lookup table $\boldsymbol{Q}^{\mathrm{S}}$ is updated with the set $\{(h(s_t), d_t)\}_{t=0}^T$ based on the following rules:

$$
Q^{\mathrm{S}}(h(s_t)) \leftarrow
\begin{cases}
(h(s_t), d_t), & \text{if } Q^{\mathrm{S}}(h(s_t)) \notin \boldsymbol{Q}^{\mathrm{S}}; \\
(h(s_t), e(h(s_t))), & \text{if } Q^{\mathrm{S}}(h(s_t)) \in \boldsymbol{Q}^{\mathrm{S}} \text{ and } e(h(s_t)) \geq d_t; \\
(h(s_t), d_t), & \text{if } Q^{\mathrm{S}}(h(s_t)) \in \boldsymbol{Q}^{\mathrm{S}} \text{ and } e(h(s_t)) < d_t.
\end{cases}
$$

The lookup operator can implicitly merge multiple trajectories, allowing the lookup table to store the historical optimal discounted return for each $h(s)$. Furthermore, researchers have proposed a class of parametric EM mechanisms (Hu et al., 2021; Ma et al., 2022). However, these parametric EM mechanisms are inherently limited by their reliance on Q-value functions. To the best of our knowledge, SfBC (Chen et al., 2023) is the only work that integrates both the DM and the EM mechanism, but it does not utilize the EM mechanism to control DMs. Hence, our work is the first to introduce the EM mechanism as a control mechanism for DMs and guide the data synthesis in RL.

### A.1.1. STATE REPRESENTATION IN EM

Although state representation has been studied in RL (Schrittwieser et al., 2020; Hafner et al., 2020; Tang et al., 2017), state representation for EM mechanisms is still naive. The default state representation is random projection (RP), a classical dimensionality reduction technique (Kononenko & Kukar, 2007). Each state is typically stored as $K$ single-precision (32-bit) floating-point values, and the total number of encoded states $n_{\mathrm{RP}}$ is approximately $N$, where $N$ denotes the total number of transitions in the dataset. Hence, the storage complexity, retrieval time complexity, and construction time complexity of corresponding EM mechanisms [4] are $\mathcal{O}(32K n_{\mathrm{RP}})$, $\mathcal{O}(\log n_{\mathrm{RP}})$ and $\mathcal{O}(n_{\mathrm{RP}} \log n_{\mathrm{RP}})$, respectively. However, real-valued encoded states are unlikely to repeat (Li et al., 2023), which leads to inefficient EM updates and high storage, retrieval, and construction complexities. However, real-valued encoded states are unlikely to repeat (Li et al., 2023), which leads to inefficient EM updates and high storage, retrieval, and construction complexities. Grid-based state representation for EM mechanisms (Li et al., 2023) is proposed to address these issues, where each dimension of states is encoded by discretizing the state space into a grid-like structure. Grid-based state representation groups states that share the same characteristics into a single cluster, enabling more effective EM updates while reducing the complexities of storage, retrieval, and construction. Each state would be stored as $b_{\mathrm{Grid}}$-bit integers ($b_{\mathrm{Grid}} < 32$, typically) and the number of encoded states $n_{\mathrm{Grid}}$ would be smaller than $N$. The storage, retrieval time, and construction time complexities of corresponding EM mechanisms are $\mathcal{O}(b_{\mathrm{Grid}} K n_{\mathrm{Grid}})$, $\mathcal{O}(\log n_{\mathrm{Grid}})$ and $\mathcal{O}(n_{\mathrm{Grid}} \log n_{\mathrm{Grid}})$, respectively. Since $n_{\mathrm{Grid}} \ll N$, the storage, retrieval time, and construction time complexities of EM mechanisms with a grid-based state representation are lower than those with the RP-based state representation. However, the above state representations are both data-independent and ignore the underlying

---

[4] In this paper, we only focus on the storage complexity of encoded states in EM mechanisms and the lookup tables in EM mechanisms are implemented by the KD-tree data structure (Bentley, 1975).

---

**Algorithm 1** Training Process of EMCES

---

**Input:** an initialized diffusion model $M_\phi$, a dataset $D$, an empty lookup table $Q^{\mathrm{S}}$;
**Output:** $M_\phi, C$;
Use IsoHash (Kong & Li, 2012) to learn $h(\cdot)$ based on $D$;
**for** each transition $(s, a, r, s')$ **do**
$\quad$ Compute discounted return $d$ and hash code $h(s)$ for $s \in D$;
$\quad$ Use $(h(s), d)$ to update $Q^{\mathrm{S}}$ based on (1);
**end for**
Compute the condition set $C$ based on $Q^{\mathrm{S}}, h(\cdot)$ and (3);
Compute $\delta(x(s, a), h(\cdot))$ based on (4),$\forall x(s, a) \in D$;
Train the EM-based CDM $M_\phi$ by using $D$ and $C$;

---

**Algorithm 2** Data Synthesis Process of EMCES

---

**Input:** a condition set $C$, an EM-based CDM $M_\phi$, number of synthetic trainsitions $N_{\mathrm{EMCES}}$;
**Output:** $D_{\mathrm{EMCES}}$;
Compute the probability based on (5) for each condition $y(x(s, a), h(\cdot)) \in C$;
Sample $N_{\mathrm{EMCES}}$ conditions from $\mathcal{C}$ based on the probability;
Use the sampled conditions and $M_\phi$ to generate transitions $D_{\mathrm{EMCES}} = \{(s_i, a_i, r_i, s'_i)\}_{i=1}^{N_{\mathrm{EMCES}}}$;

---

data distribution, leading to inefficient updates in EM mechanisms. These limitations motivate us to design an effective state representation for EM mechanisms.

### A.2. Controllable Diffusion Model

To enable control over the synthesis process, we incorporate the controllable diffusion model (CDM) (Ho & Salimans, 2021) into our work. Let $\mathcal{D}(x)$ (also denoted as $\mathcal{D}_0(x)$) represents the data distribution and $\mathcal{D}_k(x_k)$ denote a series of time-dependent distributions obtained by injecting *i.i.d.* Gaussian noise to data from $\mathcal{D}_0(x)$ during the forward noising process. Specifically, at time step $k$, $\mathcal{D}_k(x_k|x) = \mathcal{N}(x, \sigma_k^2 I)$, where $\sigma_k$ is the noise level. Given a sample $x \sim \mathcal{D}(x)$, let $y(x)$ be its corresponding condition. In the backward denoising process, Karras et al. (2022) propose to train a denoiser $M_\phi(x, y; \sigma)$ with parameters $\phi$ on an L2 denoising objective, formulated as:

$$\min_\phi \mathbb{E}_{x \sim \mathcal{D}, \sigma, \epsilon \sim \mathcal{N}(0, \sigma^2 I)} \|M_\phi(x + \epsilon, y(x); \sigma) - x\|_2^2. \tag{6}$$

The design of the condition $y(x)$ is critical (Zhang et al., 2023), as a well-designed $y(x)$ can ensure the generated data aligns with the desired features or structure. Conversely, a poorly designed $y(x)$ might cause the model to generate irrelevant or poor-fidelity outputs. Furthermore, for details on the training and sampling processes, we refer readers to the original paper (Karras et al., 2022).

## B. Algorithms of EMCES

EMCES typically has two processes: the training and data synthesis processes. During the training process, we obtain a hashing-based state representation and then construct the EM mechanism based on the existing dataset. The training process is summarized in Algorithm 1. During the data synthesis process, we prioritize sampling conditions based on EMTD-errors to guide the EM-based CDM in synthesizing high-quality data $D_{\mathrm{EMCES}}$. The data synthesis process is summarized in Algorithm 2.

## C. Hyperparameters

We list all the related hyperparameters here. In addition, we will release our code upon acceptance. The global parameters encompass the architecture and optimization settings for the denoising network, as well as the configuration of the diffusion process. The values of these global parameters are shown in Table 5 and applied in all environments and methods unless otherwise specified.

*Table 5.* Hyperparameter settings for the denoising network, optimizer, and diffusion process in all methods.

| Component | Hyperparameter | Value |
|---|---|---|
| MLP Denoising Network | number of layers | 6 |
| | width | 1024 |
| | activation function | ReLU |
| Optimizer | optimizer | Adam |
| | learning rate | $3 \times 10^{-4}$ |
| | learning rate schedule | Cosine annealing |
| | training steps | 100K |
| | batch size | 256 for online and medium-replay, 1024 otherwise |
| Diffusion Process | diffusion steps | 128 |
| | $\sigma_{\min}, \sigma_{\max}$ | [0.002, 80] |
| | $S_{\mathrm{tmin}}, S_{\mathrm{tmax}}$ | [0.05, 50] |
| | $S_{\mathrm{noise}}$ | 1.003 |
| | $S_{\mathrm{churn}}$ | 80 |

*Table 6.* Hyperparameter settings of $K$ and $\beta$ for different environments from D4RL. Please note that $K$ represents the dimension of the state space and $K \leq F$.

| Environment ($F$) | Task | Hyperparameters | |
|---|---|---|---|
| | | $K$ | $\beta$ |
| HalfCheetah (17) | random | 8 | 1e-4 |
| | medrep | 8 | 1e-2 |
| | medium | 8 | 1e-2 |
| | medexp | 16 | 1e-4 |
| Walker2d (17) | random | 8 | 1e-4 |
| | medrep | 8 | 1e-2 |
| | medium | 8 | 1e-2 |
| | medexp | 8 | 1e-2 |
| Hopper (11) | random | 11 | 1e-4 |
| | medrep | 11 | 1e-4 |
| | medium | 11 | 1e-4 |
| | medexp | 11 | 1e-4 |
| Maze2d (4) | umaze | 4 | 1e-2 |
| | medium | 4 | 1e-4 |
| | large | 4 | 1e-4 |

Furthermore, in our method, two key environment-specific hyperparameters are introduced: the dimension of the encoded state, $K$, and the weighting coefficient, $\beta$, for the EM-prioritized condition sampling strategy. $K$ controls the size of the compressed representation stored in EM. $\beta \in [0, 1]$ determines the strength of prioritization based on EMTD-errors in the EM-prioritized condition sampling strategy; $\beta = 0$ corresponds to uniform sampling, while larger values give higher importance to transitions with larger EMTD-error. Since $K \leq F$ and $F$ varies across environments (as shown in Table 6), we empirically select $K$ from $\{8, 16\}$ for HalfCheetah and Walker2d and select $K$ from $\{8, 11\}$ for Hopper. As for Maze2d, its dimension of state space $F$ is 4, so we choose $K = 4$. We also empirically select $\beta$ from $\{1e-2, 1e-4\}$ for each task. The values used in all D4RL experiments are summarized in Table 6. And we maintain consistency in hyperparameters across all tasks unless otherwise specified. As for the VD4RL environments, the dimensions of the state space $F$ are 24 and 12 for DrQ+BC and BC, respectively. Hence, $K$ is set to 16 and 8 for DrQ+BC and BC, respectively. And $\beta$ is set to 1e-4. As for the online RL setting, we set $K = 6$ and $\beta = 1e-2$.

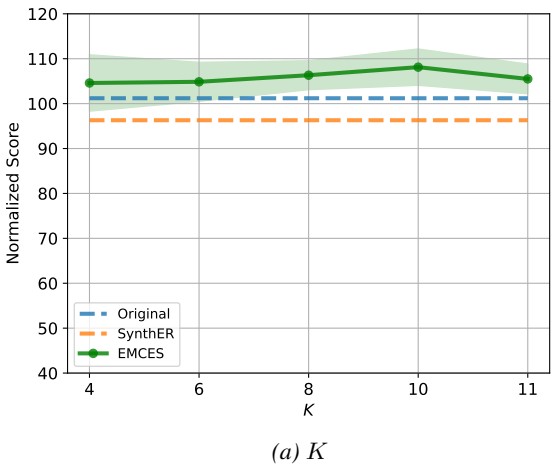

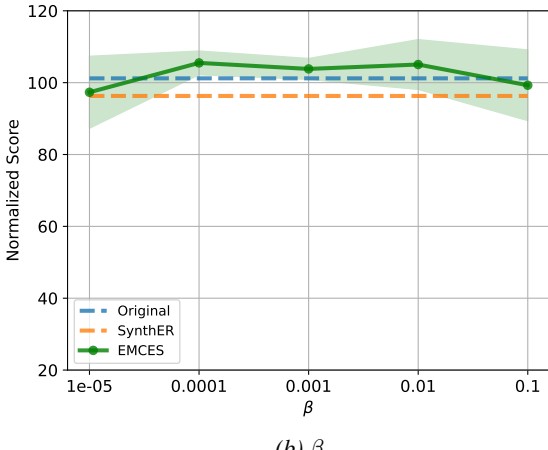

*(a) K*  *(b) β*

*Figure 5.* Sensitivity analysis of EMCES on Hopper with the 'medexp' dataset. Left: performance with different $K$. Right: performance with different $\beta$.

### C.1. Sensitivity to Hyper-parameters

To investigate the effect of hyperparameters, we choose the Hopper with the 'medexp' dataset as the evaluation task and TD3+BC as the RL algorithm. In EMCES, the hyperparameters include the length of hash codes $K$ and the priority level of the EMTD-error $\beta$. $K$ affects the granularity of state representations in the embedding space, and a larger value of $K$ enforces finer distinctions between similar states. For Hopper with the 'medexp' dataset, where the state space has a dimension of 11, $K$ is chosen from $\{4, 6, 8, 10, 11\}$ when $\beta = 1\text{e-}4$. Figure 5a presents the results of different values. We can observe that EMCES performs well when $K$ is chosen from $\{8, 10, 11\}$. The consistent performance of EMCES across $K \in \{8, 10, 11\}$ suggests that our method is relatively insensitive to the hyperparameter $K$.

$\beta$ decides the priority level of the EMTD-errors in the data synthesis process. A larger $\beta$ would tend to guide the synthesis of transitions with higher EMTD-errors. As $\beta$ approaches 0, the sampling process gradually converges to uniform sampling. Performance comparison across different values of $\beta$ with fixed $K = 11$ on Hopper is shown in Figure 5b, where $\beta \in \{1\text{e-}5, 1\text{e-}4, 1\text{e-}3, 1\text{e-}2, 1\text{e-}1\}$. We can find that our method performs better than baselines when $\beta \in \{1\text{e-}4, 1\text{e-}3, 1\text{e-}2\}$. These results verify that our method can improve the quality of synthetic data and be relatively insensitive to the hyperparameter $\beta$.

## D. Results in Offline and Online RL

### D.1. Offline RL

As the main paper claimed, we also compare our method, EMCES, with RTDiff (Yang & Wang, 2025) in offline RL. For the trajectory-based data augmentation method, RTDiff, we follow the original paper and set the width of the MLP denoising network to 4096 and the training steps to $10^6$. We performed a coarse search of hyperparameters for length from $\{2, 3, 5, 10\}$ and found that when length is set to 3, it is a reasonable choice in most environments. For Hopper and Halfcheetah environments with the 'medexp' dataset, we found that performance is best when the length is set to 2. The results of all data augmentation methods on D4RL environments are shown in Table 7. From Table 7, we can observe that our method outperforms all baselines in most environments. This verifies that our method can effectively improve the quality of the synthetic data.

### D.2. Online RL

#### D.2.1. TRAINING DETAILS IN ONLINE RL

In online RL, we use SAC (Haarnoja et al., 2018) as the RL algorithm to evaluate data augmentation methods, as it is a classical online RL algorithm. SAC sets the update-to-data (UTD) ratio to 1 (1 update for each transition collected). We also

*Table 7.* Overall results on D4RL environments using selected state-of-the-art offline RL algorithms. Note that for each task, all methods are evaluated on datasets of equal size.

| Environ-ment | Task | TD3+BC | | | IQL | | | EDAC | | |
|---|---|---|---|---|---|---|---|---|---|---|
| | | RTDiff | SynthER | EMCES | RTDiff | SynthER | EMCES | RTDiff | SynthER | EMCES |
| Half Cheetah | random | $9.9_{\pm0.5}$ | $11.5_{\pm0.6}$ | $12.5_{\pm0.4}$ | $6.4_{\pm4.2}$ | $13.8_{\pm1.2}$ | $13.7_{\pm1.8}$ | $11.9_{\pm5.0}$ | $16.5_{\pm5.9}$ | $18.6_{\pm1.2}$ |
| | medrep | $41.8_{\pm0.5}$ | $44.5_{\pm0.7}$ | $45.2_{\pm0.6}$ | $43.6_{\pm0.4}$ | $44.3_{\pm0.3}$ | $44.3_{\pm0.2}$ | $48.5_{\pm0.7}$ | $55.8_{\pm2.4}$ | $56.5_{\pm0.5}$ |
| | medium | $46.3_{\pm0.1}$ | $48.7_{\pm0.3}$ | $48.8_{\pm0.2}$ | $50.5_{\pm0.5}$ | $48.9_{\pm0.3}$ | $49.3_{\pm0.1}$ | $56.4_{\pm0.9}$ | $63.9_{\pm2.5}$ | $68.2_{\pm1.0}$ |
| | medexp | $93.4_{\pm1.3}$ | $76.2_{\pm13.4}$ | $90.7_{\pm3.3}$ | $95.6_{\pm0.4}$ | $92.1_{\pm5.2}$ | $93.8_{\pm1.2}$ | $83.4_{\pm3.4}$ | $63.7_{\pm7.6}$ | $75.8_{\pm10.0}$ |
| Walker2d | random | $1.6_{\pm1.4}$ | $1.8_{\pm2.4}$ | $4.0_{\pm2.5}$ | $5.3_{\pm0.7}$ | $3.4_{\pm1.1}$ | $3.7_{\pm0.3}$ | $5.1_{\pm7.8}$ | $10.8_{\pm8.8}$ | $21.6_{\pm1.1}$ |
| | medrep | $67.7_{\pm12.2}$ | $85.9_{\pm1.8}$ | $85.7_{\pm2.3}$ | $77.2_{\pm5.8}$ | $82.0_{\pm8.8}$ | $82.4_{\pm2.1}$ | $80.2_{\pm0.6}$ | $74.9_{\pm1.4}$ | $72.6_{\pm0.3}$ |
| | medium | $82.2_{\pm1.8}$ | $84.1_{\pm0.3}$ | $84.2_{\pm1.1}$ | $64.2_{\pm25.7}$ | $83.0_{\pm4.0}$ | $86.1_{\pm2.1}$ | $28.0_{\pm16.8}$ | $88.2_{\pm1.0}$ | $89.4_{\pm1.0}$ |
| | medexp | $110.4_{\pm0.3}$ | $110.5_{\pm0.5}$ | $110.7_{\pm0.3}$ | $112.3_{\pm0.4}$ | $111.2_{\pm1.0}$ | $111.7_{\pm0.5}$ | $109.6_{\pm0.6}$ | $111.6_{\pm1.0}$ | $112.9_{\pm0.6}$ |
| Hopper | random | $8.4_{\pm0.1}$ | $7.8_{\pm0.6}$ | $14.1_{\pm8.7}$ | $7.2_{\pm0.2}$ | $7.9_{\pm1.0}$ | $7.4_{\pm0.2}$ | $6.2_{\pm0.8}$ | $5.2_{\pm1.1}$ | $16.4_{\pm12.2}$ |
| | medrep | $71.7_{\pm14.4}$ | $58.9_{\pm14.0}$ | $68.5_{\pm14.1}$ | $19.3_{\pm13.6}$ | $42.3_{\pm37.9}$ | $101.0_{\pm0.9}$ | $100.6_{\pm0.6}$ | $101.0_{\pm0.7}$ | $99.6_{\pm0.8}$ |
| | medium | $60.1_{\pm5.8}$ | $57.3_{\pm2.6}$ | $61.5_{\pm2.8}$ | $63.0_{\pm2.6}$ | $65.9_{\pm2.3}$ | $67.4_{\pm2.7}$ | $102.3_{\pm0.2}$ | $100.0_{\pm4.3}$ | $100.4_{\pm1.4}$ |
| | medexp | $102.7_{\pm6.6}$ | $96.3_{\pm13.5}$ | $105.5_{\pm3.3}$ | $104.7_{\pm1.7}$ | $85.2_{\pm27.0}$ | $102.9_{\pm8.8}$ | $89.3_{\pm15.5}$ | $99.8_{\pm14.8}$ | $105.0_{\pm7.0}$ |
| **sum** | | 696.2 | 683.5 | **731.4** | 649.3 | 680.0 | **763.7** | 721.5 | 791.4 | **837.0** |
| maze2d | umaze | $37.6_{\pm9.2}$ | $37.0_{\pm26.8}$ | $41.9_{\pm3.1}$ | $40.3_{\pm1.7}$ | $40.4_{\pm2.4}$ | $40.4_{\pm1.2}$ | $89.2_{\pm7.1}$ | $92.1_{\pm11.0}$ | $92.2_{\pm3.6}$ |
| | medium | $56.0_{\pm4.6}$ | $50.8_{\pm32.8}$ | $87.7_{\pm47.3}$ | $34.4_{\pm0.9}$ | $35.0_{\pm1.7}$ | $35.1_{\pm0.7}$ | $60.4_{\pm11.3}$ | $61.7_{\pm16.1}$ | $95.6_{\pm27.9}$ |
| | large | $119.9_{\pm52.6}$ | $105.5_{\pm52.3}$ | $102.3_{\pm44.7}$ | $64.1_{\pm0.9}$ | $59.4_{\pm6.3}$ | $66.1_{\pm2.2}$ | $132.0_{\pm24.7}$ | $123.2_{\pm25.2}$ | $144.7_{\pm20.2}$ |
| **sum** | | 213.5 | 193.3 | **231.9** | 138.8 | 134.8 | **141.6** | 281.6 | 277.0 | **332.5** |

include REDQ (Chen et al., 2021), a sample-efficient variant of SAC, as a baseline with a UTD ratio of 20. SynthER (Lu et al., 2023b), PGR (Wang et al., 2025) and RTDiff (Yang & Wang, 2025) are included as baselines. These data augmentation methods retrain their diffusion model for every 10K transitions collected from the environment, generate 1M synthetic transitions, and then sample an equal mixture of original and synthetic data for training. In SAC with all data augmentation methods, the UTD ratio is set to 20. We implement PGR (Wang et al., 2025) using the code from its author [5], where the denoising network is larger than that of our method and other baselines. As for RTDiff (Yang & Wang, 2025), we also set the length to 3. For a fair comparison, the other settings of RTDiff are identical to those of our method and SynthER. We evaluate all methods over 200K online steps for DMC environments and 100K online steps for OpenAI Gym environments.

### D.2.2. RESULTS IN ONLINE RL

Figure 6 summarizes the training curves of all methods. We observe that our method, EMCES, still outperforms all baselines. This demonstrates again that the synthetic data generated by our method is of higher quality than that generated by other data augmentation methods.

Furthermore, we also compare the model size, memory usage, and training time of all methods in Table 8. Here, the experiments were conducted on the Hopper environment for 30K steps, and the diffusion model was retrained every 10K steps. Note that for PGR*, we replaced the original denoising network with a standardized lightweight architecture to align with other baselines. We conduct experiments on a workstation equipped with a 36-core, 72-thread Intel(R) Xeon(R) Gold 6240 CPU @ 2.60GHz, 377 GB RAM, and 8 NVIDIA GeForce RTX 2080 Ti GPUs. We can find that the memory usage and training time of our method do not incur extra costs.

## E. More Results

The controllable diffusion model is well-known to improve data-fidelity in the generative modeling literature (Ho & Salimans, 2021). We follow the prior work (Lu et al., 2023b) and compare the faithfulness of generated transitions to the real environment dynamics. Given a synthesized transition $(s, a, r, s')$, we roll out the action $a$ given the current state $s$ in the environment simulator to obtain the ground truth next state $s'$ and reward. We then measure the mean-squared error (MSE) between the ground truth and synthesized values. In offline RL, we compare our method with SynthER on the Hopper with 'medexp' dataset and roll out 10K transitions for each method. As shown in Figure 7, the fidelity of data generated by SynthER and EMCES is similar. Hence, the advantage of our method does not lie in generating high-fidelity transitions

---

[5] https://pgenreplay.github.io

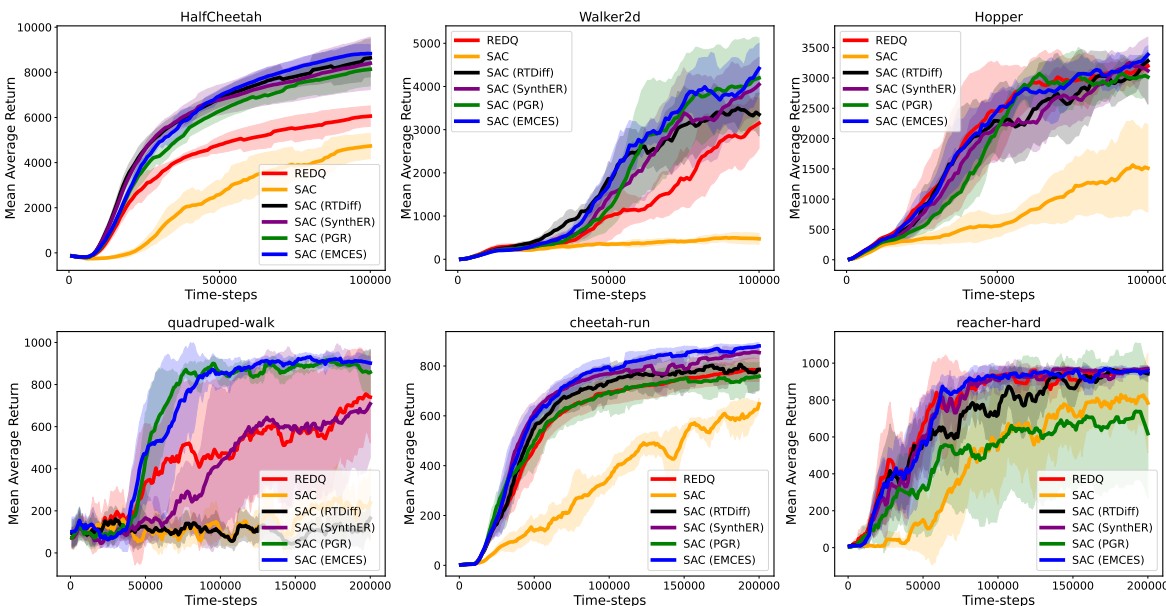

*Figure 6.* Training curves of all methods. The solid line represents the mean average return, and the shaded area represents one standard deviation.

*Table 8.* Comparison of model size, memory usage, and training efficiency among all methods in online RL.

|  | REDQ | SAC (SynthER) | SAC (PGR) | SAC (PGR*) | SAC (RTDiff) | SAC (EMCES) |
|---|---|---|---|---|---|---|
| Model Size ($\times 10^6$ params.) | 1.75 | 7.01 | 27.68 | 8.21 | 7.18 | 7.08 |
| Generation VRAM (GB) | – | 3.47 | 6.24 | 3.68 | 3.70 | 3.60 |
| Train Time, hours (Diffusion) | – | 0.83 | 1.04 | 0.90 | 0.88 | 0.84 |
| Generation Time, hours | – | 0.29 | 1.57 | 0.45 | 0.31 | 0.30 |
| Train Time, hours (RL) | 2.99 | 0.79 | 0.94 | 0.97 | 0.80 | 0.78 |
| Train Time, hours (Total) | 2.99 | 1.91 | 3.55 | 2.32 | 1.99 | 1.92 |

from the perspective of generative modeling; instead, EMCES focuses on generating high-value transitions to effectively facilitate policy learning.

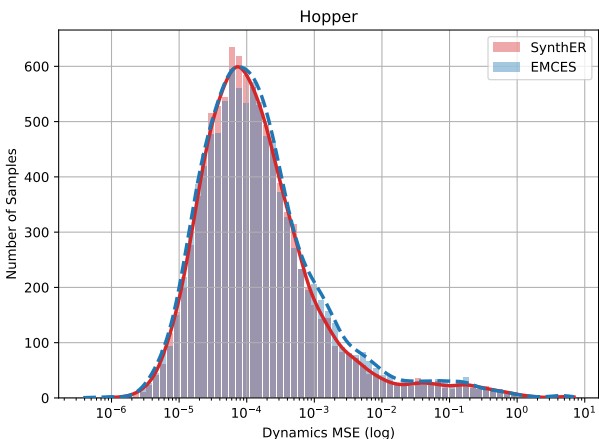

*Figure 7.* Histograms of dynamics MSE values on Hopper for SynthER and EMCES.

