# OpenReview forum: "Episodic Memory-Guided Controllable Experience Synthesis for Reinforcement Learning"
_ICML.cc/2026/Conference — ICML 2026 regular_

### Official Review · Reviewer_MzGw · 2026-02-27

**Soundness:** 3
**Presentation:** 3
**Significance:** 3
**Originality:** 3
**Overall Recommendation:** 4
**Confidence:** 3

**Summary:**

This paper addresses a critical challenge in reinforcement learning (RL)—data scarcity due to safety constraints and high interaction costs—by proposing a novel episodic memory-guided controllable experience synthesis method (EMCES). By integrating episodic memory (EM) with controllable diffusion models (CDMs), EMCES effectively improves the quality of synthetic data for both offline and online RL, overcoming the limitations of existing diffusion-based augmentation methods that lack controllability and prioritize low-informative transitions. The work is theoretically motivated, innovatively combines EM and CDMs, and is validated through comprehensive experiments across proprioceptive, vision-based, and online RL environments. The proposed components—EM-based CDM, EM-prioritized condition sampling, and hashing-based state representation—are well-designed and complementary, delivering consistent performance gains over state-of-the-art baselines. This paper makes a valuable contribution to data-efficient RL by providing a practical solution to synthesize high-quality, task-relevant data.

**Compliance With Llm Reviewing Policy:**

Affirmed.

**Final Justification:**

After considering the paper and the authors’ rebuttal, I maintain a slightly positive overall view.The paper studies a relevant problem and presents a reasonable approach with promising empirical results. The work is clearly written and generally easy to follow.While the method is sensible, the degree of innovation appears somewhat limited. The rebuttal helped clarify the differences from prior work, which I appreciate.However, I still feel that the contribution is somewhat incremental, and the paper could benefit from stronger positioning or more compelling evidence of its advantages.Overall, despite these limitations, the paper still offers useful insights and has practical value. The rebuttal improved clarity, and I am comfortable supporting acceptance.

**Key Questions For Authors:**

The paper uses a fixed β (prioritization strength) for EM-prioritized condition sampling. Could you provide a sensitivity analysis of β (e.g., testing values from 0 to 2) on 2–3 key tasks to clarify its optimal range and robustness? How would a suboptimal β affect the balance between informativeness and diversity in synthetic data?
EMCES is benchmarked against SynthER and PGR, but not against other cutting-edge generative augmentation methods (e.g., diffusion models with classifier guidance, GAN-based methods optimized for RL). Could you contextualize EMCES’s performance relative to these state-of-the-art approaches, and discuss the trade-offs between memory-guided diffusion and alternative generative paradigms?
The paper focuses on synthesizing data similar to the original dataset, but does not evaluate EMCES’s ability to generate out-of-distribution (OOD) transitions. How do you expect EMCES to perform on novel states critical for robust policy learning, and what modifications (if any) would be needed to enhance its extrapolation capability?
While EMCES is validated on VD4RL (84×84×3 images), it is not tested on larger or more complex visual inputs (e.g., 256×256 images, video sequences). How would scaling to high-resolution inputs affect state representation and synthesis quality, and what adjustments to the hashing-based state representation would be required?
The EM establishment process is deferred to the appendix. Could you provide a high-level summary of this process in the main text, and elaborate on how the choice of EM storage and retrieval strategies impacts the quality of synthetic data and downstream RL performance?

**Limitations:**

Limitations
No. The authors have not adequately discussed the limitations and potential negative societal impact of their work. Key limitations include:
Limited analysis of EMTD error sensitivity: The paper uses a fixed β (prioritization strength) for EM-prioritized sampling but does not explore how this hyperparameter affects performance across different tasks or datasets. This limits understanding of the sampling strategy’s optimal range and robustness.
Lack of comparison to recent advanced generative models: While EMCES outperforms SynthER and PGR, it does not compare to other cutting-edge generative augmentation methods (e.g., diffusion models with classifier guidance, GAN-based methods optimized for RL). This limits the contextualization of EMCES’s performance relative to the broader generative RL landscape.
Insufficient discussion of generalization to novel states: The paper focuses on synthesizing data similar to the original dataset, but does not evaluate EMCES’s ability to generate transitions for out-of-distribution (OOD) states. This restricts the assessment of its ability to extrapolate beyond the original data, which is critical for robust policy learning.
Vision-based environment scalability: EMCES is validated on VD4RL (84×84×3 images) but not on larger or more complex visual inputs (e.g., high-resolution images, video sequences). This limits claims of scalability in challenging vision-based RL settings.
Minor notation and clarity issues: Some notations (e.g.,
e(h(s))
) are introduced without immediate context, and the EM establishment process is deferred to the appendix without a high-level summary. This may hinder readability for reviewers unfamiliar with episodic memory in RL.

**Strengths And Weaknesses:**

\section*{Strengths}

\begin{enumerate}

\item \textbf{Novel Integration of Episodic Memory and Controllable Diffusion.}
EMCES is the first work to embed episodic memory (EM) into conditional diffusion models (CDMs) for RL data synthesis. This creative combination leverages EM’s ability to store high-quality past experiences and CDMs’ strength in generating realistic data. The integration directly addresses a core limitation of diffusion-based methods—lack of controllability—by using EM to construct informative and concise conditions for guided synthesis.

\item \textbf{Well-Designed Core Components.}

\begin{itemize}

\item \textbf{EM-based CDM.}
The condition design $h(s),\, r + \gamma e(h(s'))$ (encoded state and EM-estimated optimal discounted return) balances informativeness and compactness, effectively capturing transition dynamics without relying on unstable neural network-based Q-value estimation.

\item \textbf{EM-prioritized Condition Sampling.}
Using EM-based temporal-difference (EMTD) errors to guide sampling prioritizes transitions that maximize policy improvement, striking a balance between informativeness and diversity.

\item \textbf{Hashing-based State Representation.}
This data-dependent encoding reduces EM storage and computational costs while improving the quality of synthetic data, outperforming naive state representations (e.g., random projection, grid-based) by aligning with environment structure.

\end{itemize}

\item \textbf{Comprehensive Empirical Validation.}
The experiments cover diverse settings—offline RL (D4RL proprioceptive and VD4RL vision-based environments), online RL (DMC and OpenAI Gym suites)—and multiple state-of-the-art RL algorithms (TD3+BC, IQL, EDAC, SAC, DrQ+BC). EMCES consistently outperforms baselines (SynthER, PGR) and often matches or exceeds performance on real datasets, demonstrating strong generalizability and robustness.

\item \textbf{Practical Utility.}
EMCES is adaptable to both offline and online RL settings. In offline RL, it can replace or augment real data; in online RL, it supports periodic updates as new environment data is collected. The low computational overhead of its components (e.g., hashing-based representation) makes it feasible for real-world applications.

\item \textbf{Rigorous Ablation Studies.}
Detailed ablations validate the necessity of each component—condition design, EM-prioritized sampling, and hashing-based representation—confirming that each contributes to the superior quality of synthetic data and improved downstream RL performance.

\end{enumerate}


\section*{Weaknesses \& Areas for Improvement}

\begin{enumerate}

\item \textbf{Limited Analysis of EMTD Error Sensitivity.}
The paper uses a fixed $\beta$ (prioritization strength) for EM-prioritized sampling but does not explore how $\beta$ affects performance across different tasks or datasets. A sensitivity analysis (e.g., $\beta \in [0,2]$) would clarify the optimal range and robustness of the sampling strategy.

\item \textbf{Lack of Comparison to Recent Advanced Generative Models.}
While EMCES outperforms SynthER and PGR, it does not compare to other cutting-edge generative augmentation methods (e.g., diffusion models with classifier guidance, GAN-based methods optimized for RL, or hybrid approaches combining multiple generative paradigms). This limits the contextualization of EMCES’s performance within the broader generative RL landscape.

\item \textbf{Insufficient Discussion of Generalization to Novel States.}
The paper focuses on synthesizing data similar to the original dataset but does not evaluate EMCES’s ability to generate transitions for novel (out-of-distribution) states, which are critical for robust policy learning. Assessing performance under such conditions would highlight the method’s extrapolation capability.

\item \textbf{Vision-Based Environment Scalability.}
While EMCES is validated on VD4RL ($84 \times 84 \times 3$ images), it does not explore larger or more complex visual inputs (e.g., high-resolution images, video sequences) where state representation and synthesis quality become more challenging. Extending experiments to such settings would strengthen scalability claims.

\item \textbf{Minor Notation and Clarity Issues.}
Some notations lack immediate definition (e.g., $e(h(s))$ is briefly mentioned before the EM description), and the EM establishment process is deferred to the appendix without a high-level summary. A concise overview in the main text would improve readability for reviewers unfamiliar with EM in RL.

\end{enumerate}

---

> ### Author Rebuttal · Authors · 2026-03-31
>
> We sincerely thank the reviewer for the constructive comments. Below, we respond point by point.
>
> **Response to W1&Q1-2.** Thanks for your important comment. $\beta$ decides the priority level of the EMTD-errors. If $\beta$ is too small, prioritization leads to more diverse but less informative transitions; if too large, synthesis may over-focus on a narrow set with high EMTD-errors, improving informativeness but reducing diversity.
>
> We analyze $\beta\in${1e-5,1e-4,1e-3,1e-2,1e-1}, on Hopper and Halfcheetah with 'medexp' dataset using TD3+BC. The results for Hopper are shown in Fig. 5b, and those for Halfcheetah are shown below. EMCES performs best for a moderate range, $\beta\in${1e-4,1e-3,1e-2}. This suggests that EMCES is robust, and moderate prioritization balances informativeness and diversity.
>
> | |1e-5|1e-4|1e-3|1e-2|1e-1|
> |-|-|-|-|-|-|
> |Halfcheetah|85.7± 3.2|90.7±3.3|91.5±3.4|90.3±2.3|87.2±2.5|
>
> **Response to W2&Q3.** Thank you for raising this important point. We agree that broader empirical coverage would strengthen the paper. Our comparisons focus on the most closely related generative augmentation methods. As EMCES is a diffusion model (DM) based method, we primarily compare against closely related DM-based baselines.  In offline RL, besides SythER, we include RTDiff, which is a novel controllable DM-based method with state guidance. The results of RTDiff are provided in Appendix Tab. 7. In online RL, we include PGR besides SynthER and RTDiff, as shown in Fig. 6. PGR is a state-of-the-art controllable DM-based method with curiosity guidance. Thus, this submission already covers multiple representative baselines in both offline and online RL.
>
> EMCES differs from other DM-based methods by using EM to construct conditions and guide synthesis toward more policy-improving transitions. This introduces the need for an effective EM with low storage and time complexities. We address this trade-off via a hashing-based state representation, supported by the complexity analysis in Sec. 3.3.1 and empirical results in Tab. 4.
>
> **Response to W3&Q4.** Thanks for your important comment. EMCES is not designed to explicitly generate OOD transitions. Instead, it aims to synthesize high-quality transitions while remaining consistent with the underlying environment dynamics. Robust policy learning does not necessarily require extrapolation to OOD states, and it benefits from augmenting underrepresented yet task-relevant regions with plausible transitions.
>
> For moderately novel states near the support of training data, EMCES may retain some generalization ability, since its generation is based on a controllable DM with compact conditions. For states far outside the data support, its reliability is expected to decrease, as EMCES does not explicitly model uncertainty or enforce extrapolation validity.
>
> To enhance extrapolation capability, a possible direction is to incorporate model-based consistency constraints, so that synthetic data in sparse regions remain informative and dynamically plausible.
>
> **Response to W4&Q5.** Thank you for pointing this out. We agree that evaluating EMCES on higher-resolution images or video-like inputs would strengthen the scalability discussion. Following [1], we focus on the standard VD4RL setting and thus do not yet provide a systematic study in more complex visual tasks. However, EMCES is not inherently limited to this resolution. The key challenge is whether high-dimensional visual inputs can be converted into compact semantic latent representations, since hashing in EMCES operates on learned latent states rather than raw pixels. Scaling to more complex inputs would likely require stronger encoders, larger latent capacity, and possibly more expressive or temporally aware hashing schemes. For video inputs, temporal modeling would also be needed. Thus, the main challenge lies in strengthening the representation module rather than changing the EMCES framework itself.
>
> **Response to W5&Q6.** Thank you for your important comment. We agree that the presentation of EM can be clearer, especially for readers less familiar with EM in RL. In the revised paper, we will define $e(h(s))$ earlier and add a high-level summary of EM in the main text, including the two key operators of EM and its establishment process.
>
> The different EM storage and retrieval strategies differ from the choice of state representation, which determines how states from different trajectories are implicitly aggregated in EM and whether the conditions provided to the controllable DM are compact yet informative. Hence, the choice of state representation directly influences both synthetic data quality and downstream RL performance.
>
> We sincerely thank the reviewers again for insightful reviews and hope our response addresses the reviewers' concerns. We will clarify these points in the revision and would appreciate a re-evaluation in light of our response.
>
> [1] Lu, C., et al. Synthetic experience replay. In NeurIPS, 2023.

---

> > ### Author Rebuttal · Reviewer_MzGw · 2026-04-02
> >
> > I appreciate the authors' efforts in addressing my concerns

---

> > > ### Author Response · Authors · 2026-04-07
> > >
> > > Thank you for your kind acknowledgment. We sincerely appreciate your constructive comments and the opportunity to clarify our work.

---

### Official Review · Reviewer_Sp5e · 2026-03-04

**Soundness:** 3
**Presentation:** 4
**Significance:** 3
**Originality:** 3
**Overall Recommendation:** 4
**Confidence:** 3

**Summary:**

The paper points out that existing approaches that use diffusion models (DMs) to synthesize data for augmentation are still not sufficiently effective for downstream RL tasks, such that the resulting policies remain inferior to those trained with real data. To address this, the paper proposes incorporating episodic memory (EM) into controllable diffusion models (CDMs). Instead of generating transitions in an unconstrained manner, the diffusion model is guided by more informative conditions provided by EM to steer the generation toward useful directions. Moreover, EM is used to compute a criterion for which samples are more worth generating, so that the model prioritizes generating data that is more likely to improve policy learning. Experiments across multiple environments show that EMCES substantially improves the quality of synthetic data, which in turn boosts the performance of several state-of-the-art RL algorithms.

**Compliance With Llm Reviewing Policy:**

Affirmed.

**Key Questions For Authors:**

How does the hashing bit-width K trade off performance and efficiency? As K increases, does performance improve monotonically? When K becomes large enough such that collisions are rare, does the aggregation advantage of episodic memory vanish, potentially causing performance to drop?

**Limitations:**

yes

**Strengths And Weaknesses:**

**Strengths**

1. The paper’s motivation is clear and well-targeted. It explicitly demonstrates the phenomenon that current methods generate the same amount of data but achieve worse results, emphasizing that the bottleneck lies in effectiveness rather than scale.

2. Rather than simply generating more data, the method uses memory to provide value signals, leverages TD-error to prioritize the most informative samples, and employs hashing to make episodic memory practical, thereby making diffusion-synthesized experience more beneficial for RL.

3. Experimental results show that the hashed state representations substantially reduce the storage and computation overhead of EM, and the ablations reflect corresponding gains in both performance and efficiency.


**Weaknesses**

1. The paper mainly provides indirect evidence of improved synthetic data quality via higher downstream RL scores. The evaluation of generation quality relies largely on task return, and it lacks direct diagnostics of the generated transitions themselves.

2. The paper are currently centered on baselines such as SynthER and PGR. It would be more convincing to include a broader range of generative models or other types of model-based data augmentation methods.

---

> ### Author Rebuttal · Authors · 2026-03-31
>
> We sincerely thank the reviewer for their time to read our paper and provide constructive comments. Below, we respond to the raised concerns and questions point by point.
>
> **Response to Weakness 1.** Thank you for this helpful comment. Improved downstream RL performance provides evidence of higher synthetic data quality, but we agree that more direct diagnostics of the generated transitions would further strengthen the paper.
>
> We would like to clarify that our evaluation is not based solely on downstream RL performance. In the Appendix, we provide a direct transition-level diagnostic by evaluating the fidelity of synthesized transitions to the true environment dynamics. Specifically, for each synthesized transition, we rollout the state-action pair in the environment and compute the MSE between the synthesized and ground-truth next state and reward.  As shown in Fig. 8, the transition MSE remains at a low level. EMCES achieves a transition MSE comparable to SynthER, suggesting that the proposed conditioning mechanism improves downstream performance without sacrificing transition fidelity. In the revision, we will strengthen this part by explicitly presenting transition fidelity analysis as a complementary evaluation perspective.
>
> **Response to Weakness 2.** Thank you for the suggestion. We agree that a broader set of baselines would make the evaluation more convincing. Our focus is on comparing EMCES with the most closely related generative augmentation methods. In particular, diffusion models (DMs) have shown in synthesizing high-fidelity data for RL [1]. Other model-based augmentation methods are also valuable, but they are less directly comparable and can be viewed as complementary to DM-based data augmentation methods [1]. Hence, we compare against the DM-based baselines. In the offline RL setting, we compare against both SynthER and RTDiff [2], with the RTDiff results reported in Appendix Table 7 due to space constraints. In the online RL setting, our comparisons also include RTDiff in addition to SynthER and PGR, and these results are reported in Appendix Figure 6. Hence, our current submission already covers multiple representative generative baselines under both offline and online RL settings. In the revised manuscript, we will make this baseline coverage more explicit in the main text and clarify the rationale for selecting these closely related methods.
>
> **Response to Questions.** Thank you for your insightful questions. The length of hash codes (hashing bit-width) $K$ controls the granularity of the hashing-based state representation and thus induces a trade-off between state discrimination, experience aggregation, and computational efficiency. When $K$ is small, more states are mapped to the same hash code. This improves memory efficiency and strengthens aggregation across similar states, but excessive collisions may merge semantically distinct states and introduce noise into episodic memory. When $K$ is large, harmful collisions are reduced, and state discrimination improves, but the aggregation effect of episodic memory can weaken, while storage and retrieval costs increase.
>
> Therefore, we do not expect performance to improve monotonically as $K$ increases. In fact, we have conducted a sensitivity analysis over different values of $K$ on the Hopper with the 'medexp' dataset using TD3+BC. The results are shown in the Appendix, specifically in Fig. 5(a). The results show that performance first improves as overly coarse collisions are reduced, but then slightly declines as $K$ increases. Despite this, the performance of different $K$ all outperforms baselines, which verifies the effectiveness of our method. This trend is consistent with our intuition that $K$ should balance harmful collisions against beneficial aggregation, rather than be set as large as possible. We will make this trade-off more explicit in the revised paper and highlight the sensitivity results more clearly.
>
> We sincerely thank the reviewer again for the insightful review and hope that our response can address the reviewer's concerns. Meanwhile, we would greatly appreciate it if the reviewer could re-evaluate our work in light of our response.
>
> [1] Lu, C., et al. Synthetic experience replay. In NeurIPS, 2023.
>
> [2] Yang, Q. and Wang, Y. Rtdiff: Reverse trajectory synthesis via diffusion for offline reinforcement learning. In ICLR, 2025.

---

> > ### Author Rebuttal · Reviewer_Sp5e · 2026-04-03
> >
> > I appreciate the authors' efforts in addressing my concerns.

---

> > > ### Author Response · Authors · 2026-04-07
> > >
> > > Thank you for your kind acknowledgment. We sincerely appreciate your thoughtful comments and the opportunity to clarify our work.

---

### Official Review · Reviewer_TKNp · 2026-03-06

**Soundness:** 4
**Presentation:** 3
**Significance:** 3
**Originality:** 3
**Overall Recommendation:** 3
**Confidence:** 5

**Summary:**

EMCES is a data-augmentation method designed to improve the quality of synthetic transitions in Reinforcement Learning. It introduces an Episodic Memory (EM)-based Controllable Diffusion Model (CDM) that uses EM-constructed conditions to guide generation. The framework employs an EM-prioritized condition sampling strategy based on EMTD-errors, which focuses synthesis on data most beneficial for policy improvement. To maximize efficiency, a hashing-based state representation is used for EM lookup.

**Compliance With Llm Reviewing Policy:**

Affirmed.

**Key Questions For Authors:**

* Hash Bit-Length Sensitivity: How does the performance and collision rate of the hashing-based representation vary with the length $K$ of the hash codes? Is there an optimal $K$ for high-dimensional image states?
* Condition Compactness: In Equation 2, you incorporate $h(s)$ and $r + \gamma e(h(s'))$. Have you analyzed the information bottleneck here? Could adding action embeddings $h(a)$ further improve the controllability of the transition synthesis?
* EM Query Latency: For the KD-tree lookup in EM, what is the query latency impact when scaling to millions of transitions? Does the construction time $\mathcal{O}(n \log n)$ become a bottleneck in online RL?

**Limitations:**

The non-parametric EM lookup can still suffer from memory bloat in extremely large-scale tasks. The method's guiding signal is also dependent on the "quality" of experiences already present in the memory.

**Strengths And Weaknesses:**

* Strengths
  * Controllability: Successfully addresses the lack of synthesis control in existing methods like SynthER.
  * Efficiency & Complexity: The hashing mechanism dramatically reduces storage and time costs compared to random projection while achieving better performance.
  * Cross-Setting Versatility: Demonstrates strong performance in both offline and online RL, and scales effectively to vision-based environments (VD4RL).

* Weaknesses
  * Heuristic Nature of EMTD: EMTD-error is a clever proxy, but prioritizing high-error transitions could lead to a synthetic dataset that over-represents "difficult" or noisy regions at the expense of "routine" data diversity.
  * Hashing Resolution: In vision-based tasks, a learning-to-hash approach might collapse semantically distinct visual states if the bit-length $K$ is too small, potentially introducing dynamic inconsistencies.

---

> ### Author Rebuttal · Authors · 2026-03-31
>
> We appreciate your valuable time on our submission and the insightful comments. Below, we respond to the comments point by point.
>
> **Response to W1.** We agree that naively oversampling high EMTD-error transitions would lose data diversity. However, our method does not deterministically select only high-EMTD transitions.  Instead, we propose an EM-prioritized condition sampling strategy (Lines 247–262) with $\beta\in[0,1]$ to balance informativeness and diversity.
>
> We also provide a sensitivity analysis of $\beta$ in Appendix Sec. C.1 on Hopper with the 'medexp' dataset using TD3+BC. $\beta$ is chosen from {1e-5,1e-4,1e-3,1e-2,1e-1}. As shown in Fig. 5b, our method outperforms the baselines for $\beta\in${1e-4,1e-3,1e-2}. This suggests that moderate prioritization improves informativeness without compromising diversity.
>
> **Response to W2.** We agree that the hash bit-length $K$ is a key design choice in vision-based tasks. If it is too small, semantically distinct visual states may collide, leading to overly coarse aggregation.
> Following [1], for vision-based tasks, we generate transitions in a latent space rather than in the raw pixel space. Hence, we apply hashing to a compact learned latent space, which already reduces low-level visual variation. Empirically, our method is effective on VD4RL tasks, and we report the analysis of $K$ on the Cheetah_run task in response to Q1. This suggests harmful semantic collapse is not severe under the reasonable $K$ in our setting.
>
> **Response to Q1.** We analyze sensitivity to the length of the hash codes $K$ on the Hopper with the 'medexp' dataset using TD3+BC in the Appendix Sec. C.1. Here, $K\in${4,6,8,10,11}. The results show that our method performs well when $K\in${8,10,11}. We also report the number of distinct hash codes for different $K$, namely {470,489,570,690,714}, which serves as an indirect indicator of collision level: fewer distinct hash codes imply more collisions and thus coarser aggregation. For vision-based tasks, following [1], we operate in a latent space rather than raw pixel space, and we further evaluate $K\in${4,8,10,12} on cheetah_run with ‘medium’ dataset using BC. The results are summarized in the following table. This suggests that, for the vision-based task, a moderate $K$ is sufficient, and increasing $K$ does not yield monotonic gains.
>
> |$K$|4|8|10|12|
> |-|-|-|-|-|
> |Normalized Score|516.63±14.14|522.79±4.00|523.15±3.90|523.04 ±4.77|
> |Number of Hash codes|16|254|809|1901|
>
> **Response to Q2.** Our goal is to use a compact yet transition-relevant condition $(h(s),r+\gamma e(h(s′)))$, rather than a complete description of the transition. Specifically, $h(s)$ is the compact state encoding, while $r+\gamma e(h(s′))$ is the EM-estimated optimal discounted return that implicitly reflects the current action by $r=R(s,a)$ and captures the next state's quality through $e(h(s'))$.  Hence,  the condition preserves the most relevant information for controllable synthesis without making it excessively high-dimensional.
>
> We also study two explicit action variants, $a$ and $h(a)$, where $h(a)$ is obtained by IsoHash. On Hopper with the 'medexp' dataset, the results of all condition types using TD3+BC are presented below. Our method outperforms others, even with explicit action information. This suggests that explicitly adding $a$ or $h(a)$ does not provide additional benefit in our setting.
>
> |Condition Type|Normalized Score|
> |-|-|
> |$h(s),r+\gamma e(h(s'))$(Ours) |**105.5±3.3**|
> |$r+\gamma e(h(s'))$ |91.0±15.9|
> |$h(s)$|91.7±11.5|
> |$h(s),h(a),r+\gamma e(h(s'))$|101.4±9.2|
> |$h(s),a,r+\gamma e(h(s'))$ |104.8±6.6|
>
> **Response to Q3.** In EMCES, EM lookup is performed in a hashing-based state representation space rather than the original state space, which substantially reduces retrieval cost. Moreover, the size of EM is controlled by the number of hash codes, rather than growing linearly with the number of replay-buffer transitions. For example, on Hopper, the maximum number of hash codes is bounded by $2^{11}=2048$. We construct a KD-tree with all 2048 entries populated and perform $10^6$ queries: the total query time is 119.42s. The construction time is 1.7588 ms. Compared with the overall online RL pipeline (about 0.78 hours for 30K steps), this overhead is negligible. Even with millions of transitions and queries, EM retrieval remains efficient, and KD-tree construction time does not become a bottleneck.
>
> **Response to Limitations** The memory bloat of EM can be alleviated by the hashing-based state representation. By aggregating information across trajectories, EM distills the source dataset into a higher-quality memory signal. Hence, the guidance used in EMCES can be more informative than the raw dataset itself.
>
> We sincerely thank the reviewers again for constructive feedback. We will clarify these points in the revision and would appreciate a re-evaluation in light of our response.
>
> [1] Lu, C., et al. Synthetic experience replay. In NeurIPS, 2023.

---

> > ### Author Rebuttal · Reviewer_TKNp · 2026-04-04
> >
> > I'm fully appreciate that the author resolved my concern. I'll raise the score to 4.

---

> > > ### Author Response · Authors · 2026-04-07
> > >
> > > Thank you very much for your encouraging feedback. We are delighted to know that our response has addressed your concern, and we sincerely appreciate your time, consideration, and support.

---

### Official Review · Reviewer_oFMc · 2026-03-13

**Soundness:** 3
**Presentation:** 3
**Significance:** 3
**Originality:** 3
**Overall Recommendation:** 4
**Confidence:** 3

**Summary:**

The paper aims to address an existing problem in reinforcement learning: collecting high-quality offline data is costly, while existing data synthesis methods do not effectively improve policy learning. To tackle this issue, the authors propose an Episodic Memory-guided Controllable Experience Synthesizer (EMCES). The method uses a table-like episodic memory (EM) to store the historically best discounted return starting from each state. Based on this memory, the authors train an EM-based controllable diffusion model (CDM), which conditions the generation process on signals related to the estimated optimal return. Experimental results show that EMCES can improve offline RL performance using the generated synthetic data.

**Compliance With Llm Reviewing Policy:**

Affirmed.

**Final Justification:**

The authors provide helpful clarifications and additional empirical results that support the effectiveness of the proposed method. Although I still think the paper would benefit from stronger theoretical analysis, the current contribution is reasonable. I lean toward a weak accept, although I would not oppose a rejection.

**Key Questions For Authors:**

Please refer to the Weaknesses section above.

**Limitations:**

The authors discuss the limitations of existing methods that motivate their work, but they do not discuss the limitations of their own approach. For example, it would be helpful to discuss whether the method can be practically applied in real scenarios, and if not, what factors may hinder its practical adoption.

**Strengths And Weaknesses:**

Strengths:

- The idea of using the estimated return as a conditional signal for the diffusion model is interesting and provides a way to guide the generation process toward more informative transitions.

- The paper provides sufficient implementation details, and the code is available, which improves reproducibility.

- The paper includes experiments across multiple environments and RL algorithms, which helps demonstrate the effectiveness of the proposed method.

Weaknesses:

- The correctness and effectiveness of the proposed EMTD-error are not theoretically justified. Providing theoretical analysis or stronger empirical evidence could help clarify why this signal is appropriate for guiding data synthesis.

- Although conditioning the diffusion model on estimated return is an interesting idea, there is no guarantee that the generated data still follows the true environment transition distribution. This potential distribution mismatch is not discussed.

- The paper introduces a hashing-based state representation, but the analysis of its effectiveness is somewhat limited. In particular, more detailed analysis of its impact on performance and computational cost would strengthen the paper.

---

> ### Author Rebuttal · Authors · 2026-03-31
>
> We sincerely thank the reviewer for the insightful comments and the support of our work. We would like to respond to the raised weaknesses point by point below.
>
> **Response to Weakness 1.** We agree that the rationale for EMTD-error should be clarified more explicitly. EMTD-error is defined as $\delta(x(s, a),h(\cdot)) = r + \gamma e(h(s^{\prime})) - e(h(s))$. Here, $e(h(s))$ is the historical optimal discounted return retrieved from EM, viewed as the historical baseline for $s$. $ r + \gamma e(h(s^{\prime})) $ can be viewed as an EM-based one-step lookahead target. Thus, EMTD-error measures whether a transition suggests a better continuation than previously observed from the current state, and can be viewed as an EM-based analogue of the TD error. While we do not claim a formal theoretical proof in the current submission, this explains why EMTD-error is a meaningful criterion for prioritizing transitions during data synthesis.
> Empirically, Table 3 already shows that replacing EMTD-error with $e(h(s))$ or uniform degrades performance.  We further compare with return- and reward-based priority on Hopper with the 'medexp' dataset using TD3+BC, shown in the following table. Among all priority strategies, EMTD-error performs best. We will clarify this rationale and add the stronger evidence in the final version.
>
> | Priority strategy | Normalized Score |
> | -- | -- |
> | EMTD-error  | **105.5 ± 3.3**  |
> | $e(h(s))$  | 103.6 ± 6.1  |
> | Uniform | 91.6 ± 7.5 |
> | Return   | 95.2 ± 5.2 |
> | Reward   | 99.4 ± 9.8 |
>
> **Response to Weakness 2.**  Thanks for this important concern. We agree that conditioning a diffusion model on estimated return does not provide a strict theoretical guarantee that generated transitions exactly follow the true environment transition distribution. This issue is discussed empirically in the Appendix (Section E), though we agree that it should be stated more clearly.
>
> To access potential distribution mismatch empirically, following [1], we evaluate the fidelity of generated transitions to the true dynamics: for a synthesized transition $(s,a,r,s^{\prime})$, we roll out $a$ from $s$ in the environment, obtain the ground-truth next state and reward, and then compute the mean-squared error (MSE) between the synthesized and true values. A lower MSE means better consistency with the environment transition distribution.
> As shown in Fig. 7 of the Appendix, the MSEs over 10K synthesized transitions are concentrated at very small values, and EMCES achieves a similar level of dynamics MSE to SynthER. This suggests that conditioning on estimated returns does not noticeably exacerbate the distribution mismatch problem in our setting, while still improving downstream RL performance. We will clarify this point in the final version.
>
> **Response to Weakness 3.** We thank the reviewer for this important point. We agree that the effect of the hashing-based state representation (SR) should be analyzed more clearly.
> This analysis is included in Section 4.2, where we compare the hashing-based SR with RP-based, Grid-based SRs on Hopper with the 'medexp' dataset using TD3+BC ($K=11$).
> From a performance perspective, in addition to comparing the normalized scores under different SRs, we add a comparison of the number of encoded states, shown in the following table. Hashing-based SR uses substantially fewer encoded states than the RP-based SR, while using only slightly more than the Grid-based SR.  At the same time, the hashing-based SR achieves the best normalized score. These results suggest that hashing-based SR provides a better balance between state abstraction and discrimination, which is beneficial for the EM mechanism and further guides the high-quality data synthesis.
>
> From the efficiency perspective, we compare the storage cost and time (including retrieval and construction time) cost required for establishing the EM mechanism in the following table. The results show that the hashing-based SR achieves the best overall trade-off: it delivers the strongest performance while maintaining relatively low storage and time overhead. Although the hashing-based SR requires learning $K$ additional projection functions, this extra cost is small (0.6076s in our experiment).
>
> Overall, hashing-based SR provides the best performance-efficiency trade-off. We will make this analysis more explicit in the final version.
>
> |  | RP  | Grid | Hashing |
> | -- | --- | -- | -- |
> | Normalized Score  | 104.0 ± 8.1 | 98.2 ± 12.3 | **105.5 ± 3.3** |
> | Number of Encoded States | 1998461| **637** | **714** |
> | Storage Cost (KB) | 7806.49  | **0.86**    | **0.96** |
> | Time Cost (s)   | 7573.88| **293.12** | **297.12** |
>
> We sincerely thank the reviewers for their valuable time and their support of our work again. Meanwhile, we would greatly appreciate it if the reviewers could re-evaluate our work in light of our response.
>
> [1] Lu, C., et al. Synthetic experience replay. In NeurIPS, 2023.

---

> > ### Author Rebuttal · Reviewer_oFMc · 2026-04-04
> >
> > I appreciate the reviewer for providing a detailed explanation and additional empirical results. I acknowledge that these results help demonstrate the effectiveness and rationale of the method; however, I believe that a more thorough theoretical analysis would further strengthen the paper. That said, based on the experimental analysis, I will keep my evaluation unchanged.

---

> > > ### Author Response · Authors · 2026-04-07
> > >
> > > Thank you for your thoughtful comments and for acknowledging that our additional explanation and empirical results help support the rationale and effectiveness of EMTD-error. We agree that a more thorough theoretical analysis would further strengthen the paper. In this rebuttal, we aimed to clarify the key intuition behind EMTD-error, namely that it serves as an EM-based analogue of TD error and measures whether a transition suggests a better continuation than previously observed from the current state. Together with the additional priority ablation results, we believe this provides meaningful support for its suitability in guiding data synthesis. We appreciate this constructive suggestion and will make the motivation and rationale of EMTD-error more explicit in the final version.

---

### Decision · Program_Chairs · 2026-04-30

**Decision:**

Accept (regular)

**Comment:**

This paper introduces EMCES, a novel approach that integrates Episodic Memory (EM) with Controllable Diffusion Models (CDMs) for data augmentation in reinforcement learning. The core idea of using EM to guide the diffusion model's condition sampling is innovative, effectively addressing the limitation of existing methods that generate uninformative transitions. The hashing-based state representation for EM is a practical improvement that enhances computational efficiency. The empirical validation is solid, demonstrating consistent improvements over strong baselines like SynthER and PGR across both offline (VD4RL) and online (DMC, Gym) benchmarks, with clear ablation studies confirming the necessity of each component.

Overall, all issues raised by the reviewers have been addressed and ultimately resolved, with the post-rebuttal consensus converging toward acceptance. Reviewer TKNp gave positive feedback in the post-rebuttal comments and indicated that the score would be raised to 4.